# A New Approach Combining a Multilayer Radiative Transfer Model with an Individual-Based Forest Model: Application to Boreal Forests in Finland

Hans Henniger [1,2,*], Friedrich J. Bohn [1], Kim Schmidt [1,3] and Andreas Huth [1,2,4]

1   Helmholtz Centre of Environmental Research-UFZ, Permoserstr. 15, 04318 Leipzig, Germany;
    friedrich.bohn@ufz.de (F.J.B.); kim.schmidt@ufz.de (K.S.); andreas.huth@ufz.de (A.H.)
2   Institute for Environmental Systems Research, University of Osnabrück, Barbara Straße 12,
    49074 Osnabrück, Germany
3   Remote Sensing Centre for Earth System Research, Talstr. 35, 04103 Leipzig, Germany
4   iDiv German Centre for Integrative Biodiversity Research Halle-Jena-Leipzig, Puschstraße 4,
    04103 Leipzig, Germany
*   Correspondence: hans.henniger@ufz.de

**Abstract:** To understand forest dynamics under today's changing environmental conditions, it is important to analyze the state of forests at large scales. Forest inventories are not available for all regions, so it is important to use other additional methods, e.g., remote sensing observations. Increasingly, remotely sensed data based on optical instruments and airborne LIDAR are becoming widely available for forests. There is great potential in analyzing these measurements and gaining an understanding of forest states. In this work, we combine the new-generation radiative transfer model mScope with the individual-based forest model FORMIND to generate reflectance spectra for forests. Combining the two models allows us to account for species diversity at different height layers in the forest. We compare the generated reflectances for forest stands in Finland, in the region of North Karelia, with Sentinel-2 measurements. We investigate which level of forest representation gives the best results and explore the influence of different calculation methods of mean leaf parameters. For the majority of the forest stands, we generated good reflectances with all levels of forest representation compared to the measured reflectance. Good correlations were also found for the vegetation indices (especially NDVI with $R^2 = 0.62$). This work provides a forward modeling approach for relating forest reflectance to forest characteristics. With this tool, it is possible to analyze a large set of forest stands with corresponding reflectances. This opens up the possibility to understand how reflectance is related to succession and different forest conditions.

**Keywords:** forest model; radiative transfer; vegetation indices; individual-based; forest reflectance

## 1. Introduction

Forests play a major role in the terrestrial component of the global carbon cycle. They account for about 55% of the global above-ground carbon stock [1] and represent approximately 40% of the global terrestrial carbon sink [2,3]. Forests shape the surface of the Earth by comprising 31% of the land area [4] and they influence the energy balance by reflecting and absorbing sunlight. They are important for sustaining biodiversity and provide habitat for 70% of all faunal species [5–7]. Forests exhibit a diversity of spatial structures that can be dynamic due to natural succession, management or disturbances [8].

To monitor the state of forests, the conventional standard practice for foresters and ecologists alike has long been the measurement of forest inventories. Collecting inventories is time-consuming. However, in tropical forests, national forest inventories are often missing. Another approach to monitor forests is based on remote sensing observations, which provide relevant data at large scales. The amount of data is significantly raising with more and more Earth-observing satellite missions launched in the last ten years [9].

The spatial and temporal remote sensing observations offer the opportunity to gain a better understanding of forests with respect to their structure and dynamics. Satellite measurements vary in their resolution and coverage. Thus, for global observations, there is a trade-off between the spatial and temporal resolution of satellite (e.g., Landsat, Sentinel) and airborne products. The combined methods of remote sensing and field observations offers the opportunity to gain a better understanding of forests with respect to their structure and dynamics. However, the ecological interpretation of remote sensing observations of forests is challenging, and in many cases still in development.

One way to obtain information from remote sensing measurements concerning target vegetation variables (e.g., Leaf Area Index (LAI), species composition, productivity) is to use models that link the measured remote sensing measurements to the vegetation. Vegetation models have been successfully applied to study change in forests for nearly four decades, many of which differ in their applications. As one example, dynamic global vegetation models (e.g., ED by [10] and CLM4 by [11]), were initially developed to represent the interaction between vegetation and the global carbon cycle as stand-alone simulation models, but also to represent vegetation dynamics in the context of Earth system models, or alongside atmospheric (general circulation models), oceanic and cryospheric modeling frameworks [12]. These models focus on large-scale applications and they rely on simplifications to reduce complexity and computational demand (e.g., individual species simplified to plant functional types). They do not offer information at the individual tree level. For the analysis of forests in forestry and ecology, there has been a long tradition [13] of using individual forest models (e.g., FORMIND by [14] and LPJ-GUESS by [15]). FORMIND is able to represent the ecosystem dynamics of the forest by simulating each individual tree in a forest (forest gap model). FORMIND allows for the simulation of species-rich forests and also considers the size and age structure of the simulated tree community. At the same time, with increasing computing capacity, there is an opportunity to use these models to simulate large forest areas. Due to the simulation of single trees, they are also able to consider the heterogeneity of forest structure and dynamics.

An important component in vegetation models is solar irradiance and the competition for light between plants. One simple way to calculate the light climate is based on Lambert–Beer's law, which is often used by forest models. It describes the decreasing intensity of radiation as it passes through a medium (e.g., tree crowns), depending on the composition of the medium and the height of the layer. Radiative transfer models (RTMs) calculate the light climate in the forests in a more detailed way. They simulate the reflectance, interception, absorption and transmission of light through a canopy. Radiative transfer is influenced, e.g., by the amount of leaves, their characteristics (i.e., amount of chlorophyll and carotenoids, water content), the angle of the leaves struck by light and the angle between the leaves and the Sun. All these parameters are combined by coupled differential equations and allow for the calculation of reflectance of a forest for light of different wavelengths (between 300 nm and 2500 nm, depending on the model) including the reflectance, absorption and transmission of the leaves. Some RTMs are able to provide results for multiple canopy layers, whereas others assume a homogeneous canopy. RTMs are able to simulate the reflectance of the canopy, as it is measured by satellites. Canopy radiative transfer is one of the primary and long-relied-upon mechanisms by which models relate vegetation properties to surface reflectance as captured by remote sensing [16], as radiative transfer in combination with vegetation can be modeled at different levels of complexity. The representation of the vegetation for which the radiative transfer is calculated can range from a simple homogeneous to a detailed and heterogeneous 3D representation of the vegetation structure. The complexity of the solution of radiative transfer problems also varies [17,18] from numerical Monte Carlo ray tracing approaches (e.g., [19,20]) to analytical solutions using, e.g., four stream technology (e.g., [21]).

Some of the global vegetation models are coupled with simple RTMs to calculate reflectance for a wavelength from 300 to 2500 nm. The two-stream approximation is used

to calculate radiative transfer in CLM4.5 [22], ED2 [23] and CLM(SPA) [24]. Mostly, these models only use a few plant functional types and a low number of canopy layers.

With the new generation of RTMs (such as DART by [25] and mScope by [26]), it is possible to consider heterogeneous vegetation. The more complex the structure of the vegetation, the more computationally intensive the simulation of light reflectance and the interaction with the vegetation. The same applies to the simulation of vegetation on a global level. As mentioned, global vegetation models must make strong simplifications in order to be able to simulate large areas in appropriate timespans. Individual-based models describe forest structure in a more detailed way, but they are difficult to apply on a global scale due to the computational requirements. Nevertheless, they endorse the fundamental premise that the structure of forests represents an important factor for ecosystem dynamics that is lost in more aggregated modeling approaches [13].

Individual-based forest models in combination with the new generation of RTMs are therefore a promising approach to consider the complexity of forest structure and species. Their combination will aid in the development of a mechanistic understanding of the linkage between forest reflectance and forest properties such as structure and species diversity. The challenge is to develop an approach which is sensitive to forest structure and species diversity within the current, but ever-increasing, computational constraints both in simulating vegetation and radiative transfer, in order to allow for the analysis of huge forest simulations. Such a tool can also be used to gain a more general understanding of the relationships between reflectance and vegetation properties.

Here, we present an approach by coupling the new-generation RTM mScope with the individual-based forest model FORMIND. We enlarge the application field of mScope and investigate the calculated reflectance spectra of Boreal forests using forests in Finland as an example. Comparing the simulation output with Sentinel-2 data allows us to answer the following questions: How does the concept of forest representation (homogeneous or heterogeneous structure) influence the reflectance spectrum? Can the approach reproduce the variety of reflectance spectra in Finland? Furthermore, how well can we calculate the vegetation indices of the forests with this approach?

## 2. Materials and Methods

For coupling the individual-based forest model FORMIND and the radiative transfer model RTM mScope, we implemented mScope (in an adapted version of [26]) as an additional process in the forest model FORMIND. By using inventories for forest stands in Finland and the forest model, we were able to reconstruct these forests. In combination with the RTM, it was possible to calculate reflectance spectra for the visible and near-infrared range. We then compared the simulated reflectance with measured reflectance spectra from remote sensing observations (Sentinel-2).

To analyze possible applications, different levels of the forest complexity were analyzed and their influence on the reflection spectra was investigated. In addition, several vegetation indices were calculated and analyzed.

### 2.1. Study Site

For this study, we investigated 28 Boreal forest stands in Finland in the region of North Karelia, which are located in an area of about 150 km × 150 km (see Figure A1). The inventory data were collected for the FunDivEUROPE project (http://project.fundiveurope.eu, accessed on 23 April 2023) [27] in summer (August) in 2012 and again in 2017. Each of the 28 inventory plots had a size of 30 m × 30 m (Figure 1). The forest inventory contains information on species type, tree positions (x- and y-coordinates) and stem diameters at breast height from all trees. Information about the understory (e.g., shrubs, grasses, mosses) is not provided by the inventory. Based on stem diameter and tree species, other important forest attributes, such as tree height, crown diameter and leaf area index (LAI) are calculated by the forest model FORMIND. The investigated forest stands include as main species Picea Abies (Norway spruce), Pinus Sylvestris (Baltic pine), Betula Pendula

(silver birch) and Betula Pubescens (downy birch). Information about species richness and evenness, biomass, basal area and LAI can be found in Table A2 and Figure A5.

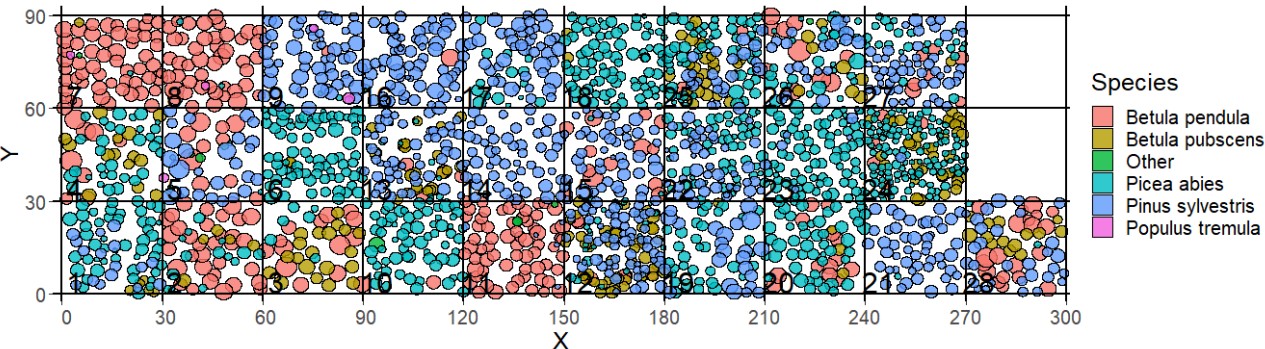

**Figure 1.** Visualization of the forest inventory of the 28 forest stands in Finland (reconstruction of 2015). Each circle represents a tree and its location in the plot (x and y coordinates). The color of the circles represents the species of a tree and the size of the circle represents its crown diameter. The number in the squares indicates the number of the forest stand and corresponds to the numbering in the FORMIND simulation. The forest stands are shown side by side but are originally distributed over an area of 150 km × 150 km (a map is shown in Appendix Figure A1).

For all forest stands, tree size was measured in 2012 and 2017. Here, we took the mean stem diameter (at breast height) of the measured stem diameter values of 2012 and 2017 as a proxy for the stem diameter in the year 2015 (same year as the analysis of Sentinel-2 data [28]) and used these values for the forest reconstruction with FORMIND. We then compared the calculated reflectance spectra with remote sensing observation using atmosphere-corrected Sentinel-2 measurements [28] from August 2015. For the simulation of the reflectance spectra, information on observation geometries (Sun and observer, in terms of zenith and azimuth) for each forest stand was provided by [28].

### 2.2. The Individual-Based Forest Model FORMIND

For the simulation of the 28 forest stands, we used the individual- and process-based forest model FORMIND, which belongs to the model family of individual-based forest gap models. This means that the growth of every single tree is simulated and that individual trees interact with each other. Additionally, FORMIND allows for the simulation of forests with different tree species and also considers the size structure of the tree community. FORMIND can be used for small-scale simulations as well as large-scale simulations [29,30], e.g., in the Amazon.

The model includes four main process groups: recruitment, mortality, competition (e.g., for light and space) and growth of each individual tree (increment of tree biomass, stem diameter and height). For our investigations, we implemented the RTM mScope as an additional process in FORMIND (in an adapted version in C++).

The stem position (x- and y-coordinate), species information and the diameter at breast height were used as input information in FORMIND. Via different allometry formulas, FORMIND calculates tree height, crown diameter and LAI. This also depends on a set of species-specific parameters and allometry equations. FORMIND has been extensively tested and applied to tropical forests [30–38], temperate forests [39–41], grasslands [42] and Boreal forests [43]. The parameterization of [39] includes all tree species of the investigated forest stands (North Karelia, Finland) and is used for our simulations on a 30 m × 30 m scale.

### 2.3. Coupling mScope with FORMIND

MScope is an RTM which, on the one hand, can handle several canopy layers, and on the other hand, has a short computation time. For this study, we [26] coupled mScope with

FORMIND (as a part of the FORMIND code). It is based on Scope (Soil Canopy Observation of Photochemistry and Energy fluxes, [44]). The Scope model is a vertical, one-dimensional, integrated radiative transfer and energy balance model, which simulates short-wave reflectance spectra (400–2500 nm) and the fluorescence of homogeneous vegetation. In its original version, it combines two basic RTMs: Fluspect [45] (on the base of PROSPECT, [46]) for calculations of reflectance, transmittance and fluorescence at leaf level and SAIL-based models (Scattering by Arbitrary Inclined Leaves, Verhoef [47]) for calculating the radiative transfer in the canopy. Compared to Scope, mScope has multiple layers to include the variation in the distribution of leaves, which enables the representation and simulation of heterogeneous vegetation.

We use mScope [26] to simulate the reflectance spectra of forest stands. For calculations at leaf level (reflectance, transmittance and fluorescence), our mScope version uses the model PROSPECT-D [46]. At canopy level (radiative transfer), a modified version of Scope is used.

For the parameterization of the leaf model, the following attributes are used:

- Leaf structure (number of internal leaf layers [layer]);
- The amount of pigments in the leaf (chlorophyll a and b [$\mu g \, cm^{-2}$], carotenoids [$\mu g \, cm^{-2}$], anthocyanins [$\mu g \, cm^{-2}$], senescent pigments [fraction]);
- Dry matter [$g \, cm^{-2}$] and leaf water content [$g \, cm^{-2}$];
- Traits describing vegetation structure as the mean and bi-modality of the leaf inclination distribution function, LAI [$m^2 \, m^{-2}$], canopy height [m].

The parameters for the different species were taken from the "CABO 2018-2019 Leaf-Level Spectra Data set" by [48] and can be found in Table A1. These values are generalized values (measurements from Finland were not available). Due to physiological similarities, the species Betula Pendula and Betula Pubescens are combined to one species group called Betula (birches). Additional information that is used is soil reflectance spectra (see Figure A2) and atmospheric constants, which are taken from [26].

*2.4. Representations of Different Levels of Forest Complexity (Heterogeneous Structure)*

Using the individual-based approach in forest modeling, it is possible to simulate and describe forest structure at fine scales, which allows for the heterogeneity of a forest to be considered. Individual-based forest models (here, FORMIND) make it possible to gain tree- and forest-specific properties for each forest patch (e.g., 30 m × 30 m) in different height layers (each height layer has a thickness/size $\Delta h$) from the bottom/soil up to the top of the canopy.

One important property to calculate radiative transfer is the LAI. FORMIND enables the calculation of LAI distributions for each tree over height (the above-described height layers). In order to determine the species composition, we used the LAI fraction of a species as a measure of its abundance. MScope uses a fixed number of height layers (in the version of [28]: 60 layers). In our modified version, we use a fixed layer height $\Delta h$ = 10 m for a low height resolution and later $\Delta h$ = 0.5 m for a high resolution (see Figure A4). We analyze forests up to forest heights of 50 m. Thus, we use 5 or 100 height layers, respectively, for our calculations. Depending on the structure of the forest, the leaves are located in different height layers. Height layers without leaves do not contribute to the reflectance spectra.

To calculate leaf reflectance and transmittance (using the leaf model PROSPECT-D), the RTM utilized information from the forest model for each layer, which included a leaf parameterization containing leaf properties for each layer. Additionally, the distribution of the orientation of leaves was considered——it was assumed to be spherical for all species——but it is also possible to choose other distributions. MScope also includes observation geometry (Sun and satellite, azimuth and zenith). Vegetation information from the reconstructed simulated forest, which is provided by the forest model, could be processed in different ways and then be transferred to the radiative transfer model. In this paper, we analyze three cases, each resulting in a different representation of the vegetation. The processing differs according to the LAI and according to the species composition (Figure 2).

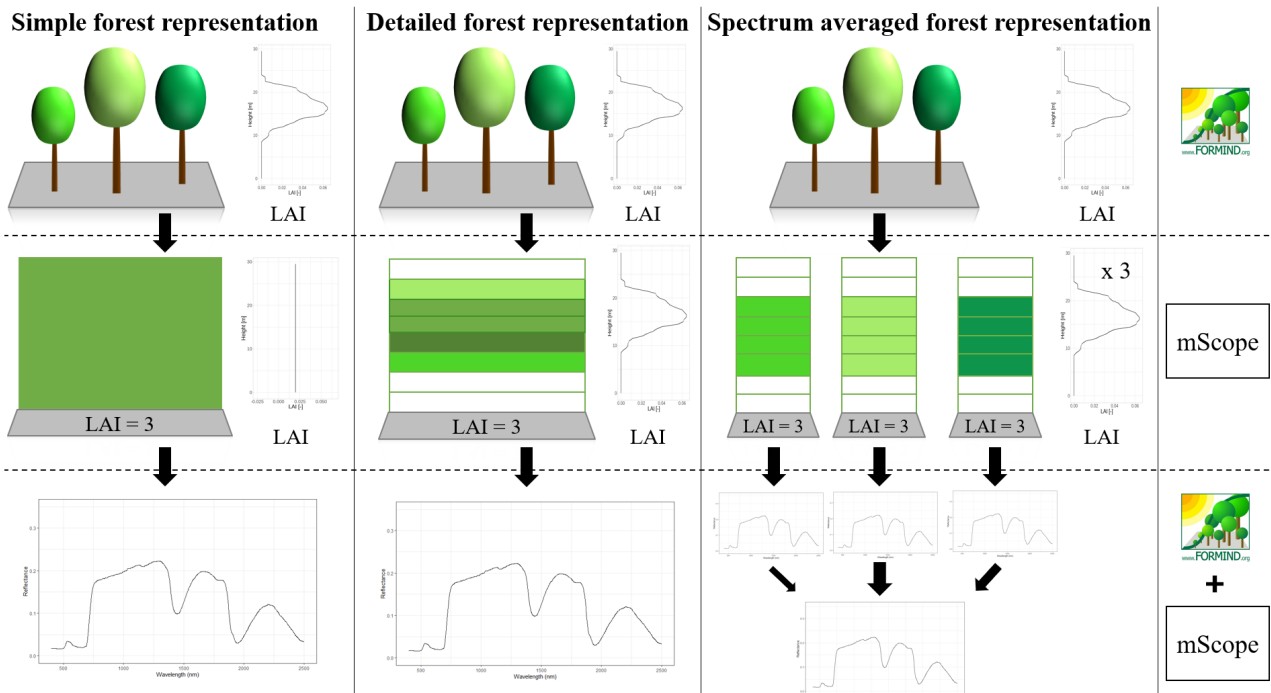

**Figure 2.** Different concepts of forest representation. Visualization of the different representations of a sample forest during the simulation (first column: simple forest representation, second column: detailed forest representation, third column: spectra-averaged forest representation). We see how, under the given concept, the forest is represented in FORMIND (first row), how it is simulated in mScope (second row) and how the output is built (third row). The sample forest has 3 different species (represented by the different colors). The concepts of representation are described in detail in the text below.

1. Simple forest representation
   The simplified forest representation only uses reduced information of the forest. It assumes the same mixture of species and the same LAI for each height layer of the forest stand. The leaf parameterization is calculated by averaging the leaf attributes of the occurring species (weighted by LAI, as a measure of abundance). The LAI of the forest stand is equally distributed among all layers.
2. Detailed forest representation
   The detailed representation of the forest assigns to each height layer different mixtures of species and different LAIs. The leaf parameterization for each layer is calculated by averaging the leaf attributes of the occurring species weighted by LAI in the height layer, as a measure of abundance. For each layer of the forest, the calculated LAI of the reconstructed forest stand will be used.
3. Spectra-averaged representation
   In this case, the forest is divided into different "sub-forests". In each sub-forest stand, we maintain the total number of trees and the structure of the main forest stand. However, we assume that all trees in a sub-forest stand are of only one species. Thus, there are as many sub-forests as there are tree species. For each layer, the calculated LAI of the reconstructed forest stand is used. For each of these single-species sub-forests, the reflectance spectra are calculated using the species-specific leaf parameters. The final reflectance spectrum is determined by averaging the species-specific spectra weighted by LAI fraction, as a measure of abundance.

The processed Sentinel-2 observations [28] include reflectance values for only 10 wavebands. MScope calculates radiative transfer for wavelengths in the range from 400 nm to 2500 nm (with a resolution of 1 nm). For better comparability with the simulated re-

flectance profiles, we averaged the simulated reflectance values for the different Sentinel-2A bands (e.g., Sentinel Band 704 nm has a range of 15 nm, so we averaged 15 reflectance values; for more information on bands see Appendix Table A3). This averaged values are shown as dots in Figures 3, 4 and A13 and are the basis of the comparisons with Sentinel-2 measurements.

Vegetation indices derived from canopy reflectance are widely used in remote sensing, as they represent proxies for vegetation attributes (e.g., LAI, productivity). We calculated several vegetation indices (NDVI, EVI, MSI, in appendix: NDMI, kNDVI). NDVI is chlorophyll-sensitive. EVI [49] is responsive to canopy structural variations, including LAI, canopy type and plant physiognomy [50]. We also analyzed kNDVI [51] as a modification of the NDVI. The NDMI is partly correlated with the water content of the canopy [52]. Ref. [53] introduced the moisture stress index (MSI, [54]), which utilizes reflectance wavebands in the SWIR (1550–1750 nm) and NIRS (760–900 nm). Additionally to the vegetation indices, we analyzed the similarity index SAD [55] (spectral angle distance see Appendix Figure A12 and Table A4).

In the mScope model, some code adjustments were made to account for the structure of the forest models and forests from the inventory. In forest models, it is possible that there are layers without leaves (vertical gaps). Adjustments were necessary to ensure that these layers had no influence on the reflectance spectrum. MScope calculates the probability of viewing a leaf in solar ($P_S$) and observer direction ($P_O$) by assuming a homogeneously distributed LAI in the forest.

$$P_S = e^{k \cdot xl \cdot LAI} \tag{1}$$

$$P_O = e^{K \cdot xl \cdot LAI} \tag{2}$$

with $xl$ as negative cumulative layer thickness, $k$ as extinction coefficient in direction of the Sun, $LAI$ as leaf area index of forest stand and $K$ as extinction coefficient in the direction of the observer.

This leads to the situation that the probability is also influenced by layers with an LAI of 0. We have changed the calculation equivalently, allowing for different LAI values for the height layers.

$$P_S = e^{-k \cdot LAI(i)} \tag{3}$$

$$P_O = e^{-K \cdot LAI(i)} \tag{4}$$

with $LAI(i)$ as leaf area index in height layer $i$ of forest stand, $k$ as extinction coefficient in the direction of the Sun and with $K$ as extinction coefficient in the direction of the observer.

The mScope code also includes a correction of $P_S$ and $P_O$, which we also considered.

## 3. Results

First, to reduce the complexity of the analysis, we analyzed the reflectance of even-aged forests, where the RTM uses a low resolution (height layer size $\Delta h$ = 10 m, Figure 3). In each layer a homogeneous leaf distribution is assumed. The even-aged forest stand number 17, which was dominated by one species, and stand number 5, which contained three species, were used as examples for this analysis. Reflectance was then calculated for simplified, detailed and spectra-averaged forest representations.

There were differences (up to 140%) in reflectance between the detailed (blue) and the simplified (orange) forest representation. The simplified representation consistently produced higher reflectance (especially in comparison to spectra-averaged representation). Both the modeling and satellite measurements show different reflectance spectra for the two forests. We found a higher similarity of reflectance for the detailed representation.

In the next part of the investigation, we increased the represented complexity of the forest by assuming a layer height of 0.5 m (Figure 4). Here, the forest model (here FOR-MIND) provided mScope with a higher resolution distribution of LAI and species-specific

information over height. As in Figure 3, the results are again shown for the simplified, detailed and the spectra-averaged representation of the forests for both example sites.

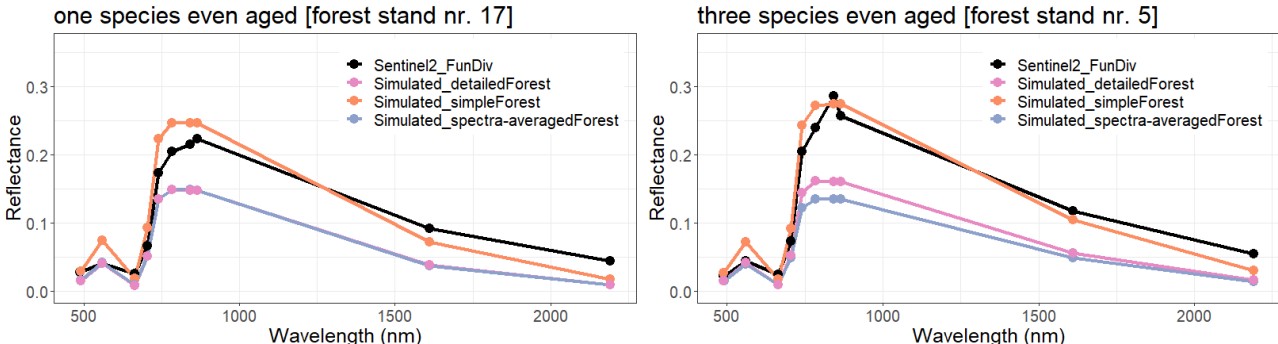

**Figure 3.** Reflectance spectra for detailed and simplified forest representation by using layers with a size of 10 m. Visualization of the calculated reflectance profiles for two forest stands (numbers 5 and 17) in Finland. Forest stand number 17 (left, mainly one species) and forest stand number 5 (right, three species). Both forest stands are even-aged (small standard deviation of tree heights: 4.6 m and 4.5 m). Each point represents the reflectance value averaged over the specific bands (corresponding to the bands of Sentinel-2). Sentinel measurements are shown in black and simulated reflectance is shown in orange/blue/pink from coupling a forest model (FORMIND) with mScope. We used 10 m height layers. The reflection of all other forest stands is shown in the Appendix (Figure A6).

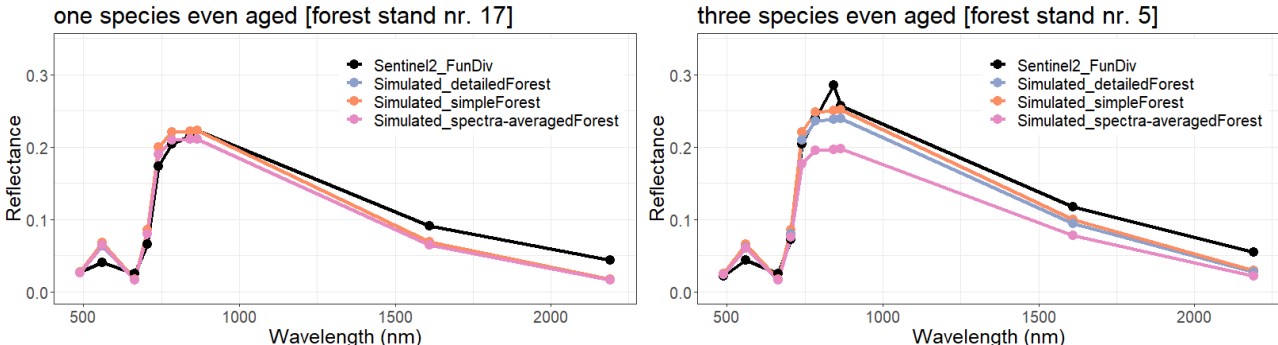

**Figure 4.** Reflectance spectra for detailed and simplified forest representation by using standard layers with a size of 0.5 m. Visualization of the calculated reflectance profiles for two forest stands (numbers 5 and 17) in Finland according to Figure 2 is shown. Each point represents the averaged reflectance value over the specific bands (corresponding to the bands of Sentinel-2). Sentinel measurements are shown in black and simulated reflectance is shown in orange/blue/pink from coupling a forest model (FORMIND) with mScope. We use here 0.5 m height layers. The reflection of all other forest stands is shown in the Appendix (Figure A10) Additionally, the reflection for the complete spectra of all other forest stands is shown in the Appendix (Figure A11). Additionally, we calculated the spectral angle distance for all comparisons (see Appendix Figure A12).

All three versions produced comparable reflectance spectra (especially for forest stand number 17). The lowest reflectances were produced with the spectra-averaged forest representation (in particular for forest stand number 5 with underestimating the NIR values). For forest stand 17, the spectra-averaged forest representation produces the same reflectance values as the detailed forest representation version. As the forest stand contains only one species, there is no averaging in the leaf parameters and spectra, and we obtained the same results for these versions. The results for all bands were in agreement with the Sentinel measurements.

The simulated reflectance spectra also enabled the calculation of vegetation indices (see Section 2.3). We analyzed NDVI, EVI and MSI (kNDVI and NDMI in Appendix

Figure A19) for each forest stand and for each forest representation (Figure 5). Each were then compared with indices calculated using the satellite observations.

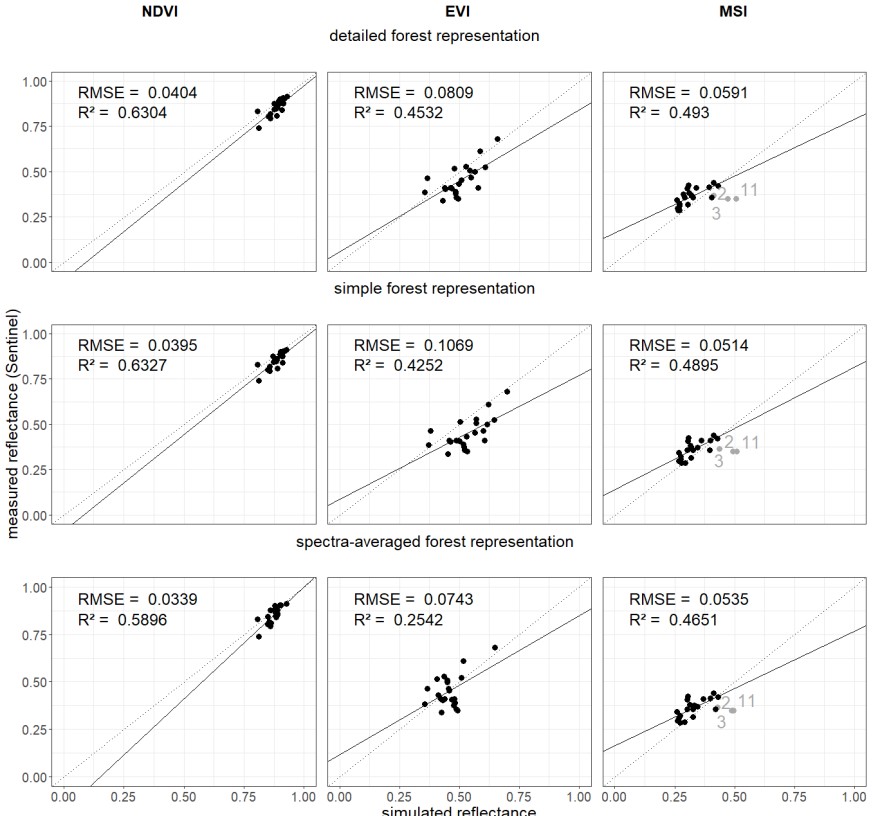

**Figure 5.** Comparison of vegetation indices. The vegetation indices (NDVI left, EVI middle, MSI right) are calculated from the reflectance values in the different wavebands once for the simulated reflectance spectra and for the satellite measurements. In each row, a different forest representation is used to calculate the results of the indices from the simulated spectra (1. detailed forest representation, 2. simple forest representation, 3. spectra-averaged forest representation; more information about the cases in Section 2.3). Each point represents a forest stand in Finland (gray points indicate birch forest stands that are not used to calculate the RMSE and $R^2$—see Appendix Figures A14–A17). Results for the calculation of the NDMI and the kNDVI can be found in the Appendix (Figure A19). We excluded five forest stands from our analysis due to inconsistencies in Sentinel-2 measurements (see Appendix Figures A14–A17).

We obtained different results for all three forest representations when analyzing NDVI, EVI and MSI. Lower $R^2$ and higher RMSE values are obtained for MSI. Measured Sentinel-2 values were close to each other. NDVI values from simulated reflectance spectra were within small ranges. We found an $R^2$ of 0.63 (detailed forest representation) when comparing simulated and measured NDVI values. For the EVI, there is a larger range of values. EVI led to a lower $R^2$ (about 0.45) and higher RMSE (0.08) compared to NDVI. The MSI of birch forest stands was overestimated (gray points). Detailed and simple forest representation show similar results for all three indices.

## 4. Discussion

In this work, we developed a new approach to study forest reflectance for radiation in the visible and near-infrared spectrum. For this, we coupled the individual-based forest model FORMIND with an adapted version of the radiative transfer model mScope. We then used the coupled models to reconstruct 28 forest stands in Finland and to calculate

reflectance spectra for each. We analyzed three different concepts of forest representation: simple, detailed and spectra-averaged.

When we compared the simulated reflectance spectra with the Sentinel measurements, the best results where achieved for the detailed forest representation. However, the measured reflectance of forests stands with similar forest structure and species mixture shows large differences in five cases (Appendix Figures A14–A17). The analysis of these cases (outliers) suggests that factors other than LAI distribution and species composition are here responsible for the differences such as limitations in the atmospheric correction or overlapping of tree crowns in the neighborhood of the forest stands. In addition to this, the approach shows potential for improvement in the sensitivity of simulated reflectance for Sentinel-2 bands B01, B03 and B04. The quality of the simulated reflectance spectrum does not depend on certain species or forest structures (we did not find general relations). This study provides a baseline for further research. The coupling of individual-based forest models and multi-layer RTMs opens up the opportunity to analyze a vast range of forests with various structure and species mixtures and to gain a deeper understanding of the reflectance spectra of complex forests (e.g., influence of tree allometries, leaf parameters or role of understory).

An important aspect of our study is the representation of the forest in the the RTM. The simple and detailed forest representations use an averaged leaf parameterization for each height layer (using the LAI of the occurring species as weighting factor). In the spectrum-averaged version, we simulated each occurring species as a monoculture forest and afterwards averaged the resulting reflectance spectra (using the LAI of the occurring species as weighting factor). Despite the non-linear nature of the RTM, the best results were obtained when the input leaf parameters were averaged (simple and detailed concept). Less satisfactory results were obtained when the output reflectance (reflectance spectra for each species) was averaged (spectra-averaged concept). We obtained similar results for simple and detailed forest representation. The NDVI values were all within a smaller range. As only a few of the analyzed stands had an LAI below 2.5, we also observed a saturation of the NDVI values [56]. Forest stands covering a broader spectrum of LAI values will allow for a more general comparison of satellite-based and modeled indices and should be conducted in future studies. For the EVI values, a lower correlation and a higher RMSE compared to the NDVI analysis was observed.

A challenge for the parameterization of radiative transfer models is the selection of suitable parameters (e.g., for leaf attributes, soil and leaf angle distribution). There are a large number of measurements available that include different leaf parameters. However, the leaf parameters of each species can vary depending on the site, the position of the leaf within the canopy, the day of the year of the measurement and environmental factors [57]. Therefore, leaf parameterizations from sites with the most comparable environmental conditions should be used. A sensitivity analysis [58–61] was used to analyze the influence of leaf parameters on the reflectance spectrum. In particular, higher sensitivity [48] is observed for those parameters that influence the visible light spectrum (e.g., pigments). Using hyperspectral data, this approach can also be used to fit species parameters.

For soil reflectance, often a wet soil type is assumed (due to a lack of data) and, in this study, we followed this approach. Nevertheless, it is also be possible to model the soil reflectance spectrum with an additional model (e.g., the BSM model by [62]).

In this study, we developed a forward modeling tool for connecting forest reflection with forest properties. There are further interesting analyses possible based on this approach. One example may be to analyze more complex forests, such as tropical forests. The information about reflectance can be used as an addition to, e.g., LIDAR measurements, to analyze forest structure and functions. It is useful to point out here that the forest model is not only able to investigate structural information but is also able to calculate characteristics of forest dynamics such as productivity. The combination of height-dependent information about forest structure with the information about light reflection spectra may give sufficient information about structure and species composition, resulting in the capability to derive,

e.g., estimates of current carbon pools. In addition to the work by [63], the presented approach makes it possible to improve the matching of satellite measurements (e.g., LI-DAR profiles) to forest simulations considering spatially heterogeneous environmental and ecological conditions. As a result, it can improve the carbon estimates for large regions. Please note that the presented approach could be used to derive simulated LIDAR profiles (and thus may improve the LIDAR model used in the mentioned study).

Importantly, this approach can also be used to generate a large number of reflectance spectra for forests by simulating forests over time and tracking reflectance spectra. This may allow us to understand the dynamics of reflectance spectra during forest succession. Disturbed forests show similar characteristics as forests in the early and mid-successional phases. We can use this knowledge to characterize disturbed forests based on reflectance. This may help us to distinguish better between natural and disturbed forests.

However, forest simulations also include path dependencies. Not all types of forest may be covered in simulations, which might occur due to management or disturbances. To overcome this, the Forest Factory approach [64,65] generates a broad range of forest states covering various types of forest structures and species compositions. This approach can also be used to identify which forests or forest states provide the same reflectance spectrum, opening up the possibility of the inversion of reflectance spectra. On the one hand, we can relate a reflectance spectrum to a set of different forest structures. On the other hand, we could also attribute a reflectance spectrum to different leaf parameters [66].

These types of studies could also be conducted for different climate scenarios, for different management strategies and regions/biomes (e.g., using the large set of available forest parameterizations for FORMIND [64,67]). Lookup tables and artificial intelligence can help us analyze such large sets of forests and their reflectance spectra and, if desired, even offer the possibility to incorporate additional information about the forests using the forest model.

## 5. Conclusions

In this work, we have applied an adapted version of the radiative transfer model mScope to a complex vegetation structure modeled by the individual-based forest model FORMIND. We showed that the weighted averaging of leaf parameters could be a useful approach to simulate reflectance of forests with different species mixtures (simple/detailed representation). The investigated types of forest representation provide good simulated reflectance spectra (for optical and NIR-range) compared to satellite measurements. However, which type of forest representation provides the best results is influenced by forest structure. In respect to vegetation indices, the best results were obtained assuming the simple or detailed forest representation. Good correlations were found between simulated and measured vegetation indices (especially NDVI). For future studies, we intend to take advantage of the detailed representation of the forest, and plan to study more heterogeneous forest stands, such as tropical forests. In combination with the forest model, many new perspectives emerge that provide the opportunity to better understand the relationship between forest reflectance and forest properties.

**Author Contributions:** Conceptualization, H.H., F.J.B. and A.H.; methodology, H.H.; software, H.H., K.S. and A.H.; formal analysis, H.H.; data curation, H.H. and K.S.; writing—original draft preparation, H.H. and A.H.; writing—review and editing, H.H., F.J.B. and A.H.; visualization, H.H.; supervision, F.J.B. and A.H.; project administration, A.H. All authors have read agreed to the published version of the manuscript.

**Funding:** The research was supported with funding from the Helmholtz Research Program "Changing Earth-Sustaining our Future"/Topic 5 "Future Landscapes" (POF-IV, 2021–2027).

**Institutional Review Board Statement:** Not applicable.

**Informed Consent Statement:** Not applicable.

**Data Availability Statement:** The code of the forest model FORMIND is available at https://formind.org (accessed on 23 April 2023). The code of the radiative transfer model mScope is available at https://github.com/peiqiyang/mSCOPE (accessed on 19 May 2023). All leaf parameters used in this study are available through the CABO data portal https://data.caboscience.org/leaf (accessed on 19 May 2023). The field data of FunDivEUROPE plots can be requested from Michael Scherer-Lorenzen (michael.scherer@biologie.unifreiburg.de).

**Acknowledgments:** We thank Javier Pacheco-Labrador for providing help regarding the code of mScope. We also thank Amanda Armstrong for providing many helpful suggestions and comments. The data from FunDivEUROPE network was collected with the financial support from the European Union Seventh Framework Programme (FP7/2007-2013) (grant agreement number: 265171). Remeasurements across all FunDivEUROPE plots were financed by EU H2020 project Soil4Europe (Bioidversa 2017-2019). The free use of Sentinel-2 data is enabled by ESA's Copernicus Open Access Hub.

**Conflicts of Interest:** The authors declare no conflict of interest. The funders had no role in the design of the study; in the collection, analyses, or interpretation of data; in the writing of the manuscript; or in the decision to publish the results.

## Abbreviations

The following abbreviations are used in this manuscript:

| | |
|---|---|
| RTM | Radiative Transfer Model |
| mScope | multilayer Soil Canopy Observation of Photochemistry and Energy fluxes |
| LAI | Leaf Area Index |
| SWIR | Short-Wave Infrared |
| NIRS | Near-Infrared Spectrum |
| RMSE | Root Mean Square Error |
| MAE | Mean Absolute Error |
| NDVI | Normalized Difference Vegetation Index |
| EVI | Enhanced Vegetation Index |
| MSI | Moisture Stress Index |
| NDMI | Normalized Difference Moisture Index |
| kNDVI | kernel NDVI |
| SAD | Spectral Angle Distance |

## Appendix A. Additional Information on the Method Section

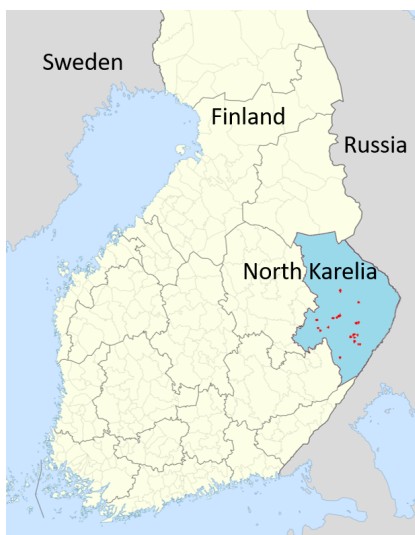

**Figure A1.** Study map of the 28 forest stands in the region North Karelia (blue area), Finland. The forest stands are distributed over an area of 150 km × 150 km.

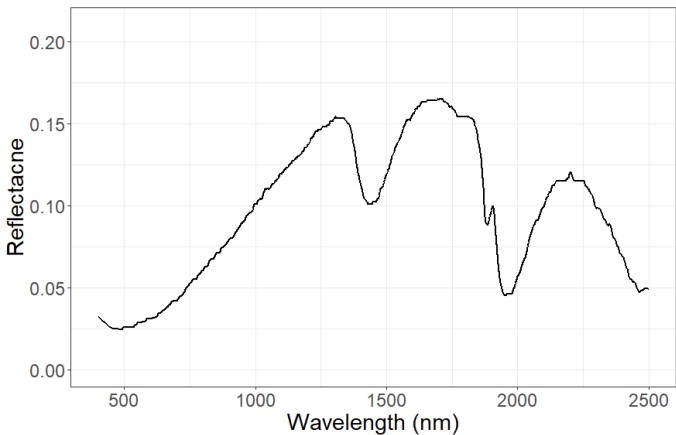

**Figure A2.** Soil reflection. Shown is assumed the reflection of wet soil. We assume for all forest stands the same soil reflectance.

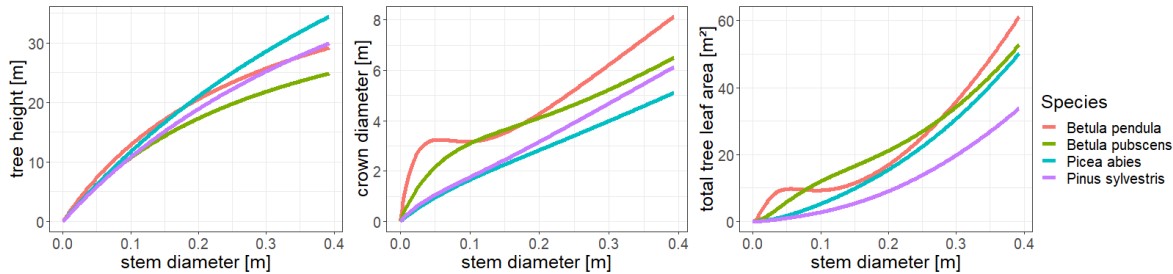

**Figure A3.** Analysis of allometries in FORMIND. Shown are the relationships between stem diameter and tree height, crown diameter as well as the total leaf area of a tree for the tree species simulated in FORMIND.

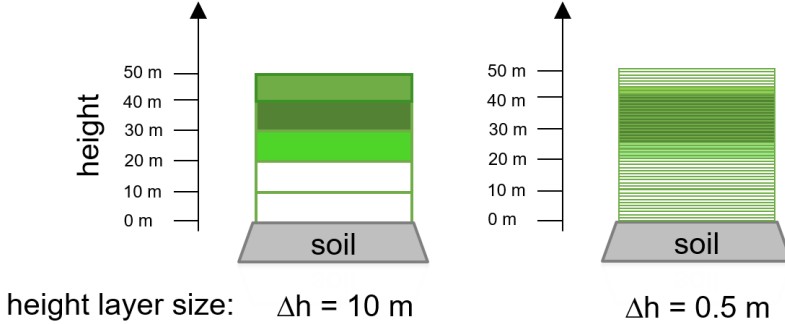

**Figure A4.** Concept of height layers. Shown is the same forest with a different resolution of height layers ($\Delta h$ = 10 m and $\Delta h$ = 0.5 m). Each layer includes different species mixtures (indicated by color). We use 5 height layers in the case $\Delta h$ = 10 m and 100 height layers in the case $\Delta h$ = 0.5 m. All layers which contain leaves contribute to the resulting reflectance spectrum. Empty height layers (no leaves are in the layer) do not influence the reflectance spectrum (details see Section 2.4).

**Table A1.** Leaf parameters.

| Leaf Parameter | Picea Abies | Pinus Silvestrys | Betula (Pendula and Pubescens) |
|---|---|---|---|
| Cab [$\mu$g cm$^{-2}$] | 21.94 | 23.92 | 36.71 |
| Cdm [g cm$^{-2}$] | 0.024 | 0.025 | 0.006 |
| Cw [g cm$^{-2}$] | 0.03 | 0.03 | 0.0117 |
| Cs [$-$] | 0.01 | 0.01 | 0.01 |
| Car [$\mu$g cm$^{-2}$] | 4.40 | 4.50 | 8.62 |
| N [$-$] | 1.25 | 1.24 | 1.77 |

**Table A2.** Attributes of the forest stands from Finland used for this study (28 plots, 30 m × 30 m).

| Plot Number [−] | Basal Area [m² ha⁻¹] | Maximum Height [m] | Height Heterogeneity [m] | Species Richness [−] | Species Evenness [−] | Biomass [$t_{odm}$ ha⁻¹] | LAI [−] |
|---|---|---|---|---|---|---|---|
| 1 | 28.78 | 30.09 | 5.31 | 3 | 0.49 | 148.30 | 3.93 |
| 2 | 19.63 | 28.63 | 3.47 | 2 | 0.35 | 106.83 | 3.23 |
| 3 | 16.31 | 26.38 | 2.88 | 2 | 0.26 | 90.74 | 2.69 |
| 4 | 22.25 | 29.21 | 4.54 | 2 | 0.41 | 118.74 | 3.46 |
| 5 | 19.87 | 29.71 | 4.64 | 5 | 0.74 | 109.47 | 2.26 |
| 6 | 32.84 | 30.80 | 4.04 | 2 | 0.04 | 170.06 | 4.84 |
| 7 | 17.60 | 23.84 | 3.36 | 4 | 0.16 | 94.34 | 3.13 |
| 8 | 17.33 | 24.59 | 2.37 | 3 | 0.17 | 95.09 | 2.94 |
| 9 | 25.66 | 26.10 | 3.69 | 5 | 0.25 | 133.24 | 2.39 |
| 10 | 27.03 | 30.01 | 4.59 | 2 | 0.05 | 139.91 | 3.98 |
| 11 | 17.02 | 22.70 | 2.92 | 3 | 0.27 | 85.02 | 3.15 |
| 12 | 27.35 | 23.32 | 3.39 | 3 | 0.65 | 114.86 | 4.07 |
| 13 | 20.38 | 21.67 | 3.28 | 3 | 0.59 | 88.96 | 2.64 |
| 14 | 18.37 | 23.34 | 2.77 | 1 | 0.00 | 86.37 | 1.67 |
| 15 | 21.37 | 26.22 | 3.38 | 3 | 0.44 | 105.38 | 2.37 |
| 16 | 27.37 | 26.44 | 4.08 | 3 | 0.10 | 139.98 | 2.52 |
| 17 | 24.54 | 28.80 | 4.52 | 2 | 0.38 | 125.61 | 2.58 |
| 18 | 30.99 | 28.45 | 4.24 | 2 | 0.07 | 152.92 | 4.71 |
| 19 | 30.12 | 28.74 | 3.61 | 3 | 0.47 | 162.74 | 3.56 |
| 20 | 30.42 | 30.30 | 3.92 | 2 | 0.35 | 164.55 | 4.52 |
| 21 | 17.60 | 23.34 | 2.45 | 2 | 0.06 | 84.49 | 1.60 |
| 22 | 27.16 | 25.85 | 3.96 | 2 | 0.43 | 120.19 | 3.54 |
| 23 | 29.31 | 27.40 | 4.08 | 2 | 0.16 | 139.35 | 4.44 |
| 24 | 25.45 | 21.70 | 2.63 | 3 | 0.38 | 89.55 | 5.04 |
| 25 | 30.72 | 27.96 | 4.61 | 3 | 0.62 | 145.02 | 4.58 |
| 26 | 26.66 | 34.41 | 5.57 | 4 | 0.69 | 138.13 | 3.67 |
| 27 | 22.03 | 22.77 | 2.95 | 3 | 0.55 | 95.38 | 2.64 |
| 28 | 22.10 | 25.73 | 3.21 | 3 | 0.50 | 114.78 | 2.97 |

Basal area is defined by the cross-sectional area of trees at breast height. Maximum height describes the highest tree height in the forest stand. Height heterogeneity describes the standard deviation of tree height. Richness describes the number of species in the forest. Evenness is defined by the normalized Shannon index. Biomass describes the sum of all tree biomass.

**Table A3.** Spectral configuration of the 10 Sentinel-2A bands used in this study [28].

| Spectral Band | Center Wavelength [nm] | Band Name | Band Width [nm] | Spatial Resolution [m] |
|---|---|---|---|---|
| B02 | 490 | blue | 65 | 10 |
| B03 | 560 | green | 35 | 10 |
| B04 | 665 | red | 30 | 10 |
| B05 | 705 | red-edge 1 | 15 | 20 |
| B06 | 740 | red-edge 2 | 15 | 20 |
| B07 | 783 | red-edge 3 | 20 | 20 |
| B08 | 842 | NIR 1 | 115 | 10 |
| B08a | 865 | NIR 2 | 20 | 20 |
| B11 | 1610 | SWIR 1 | 90 | 20 |
| B12 | 2190 | SWIR 2 | 180 | 20 |

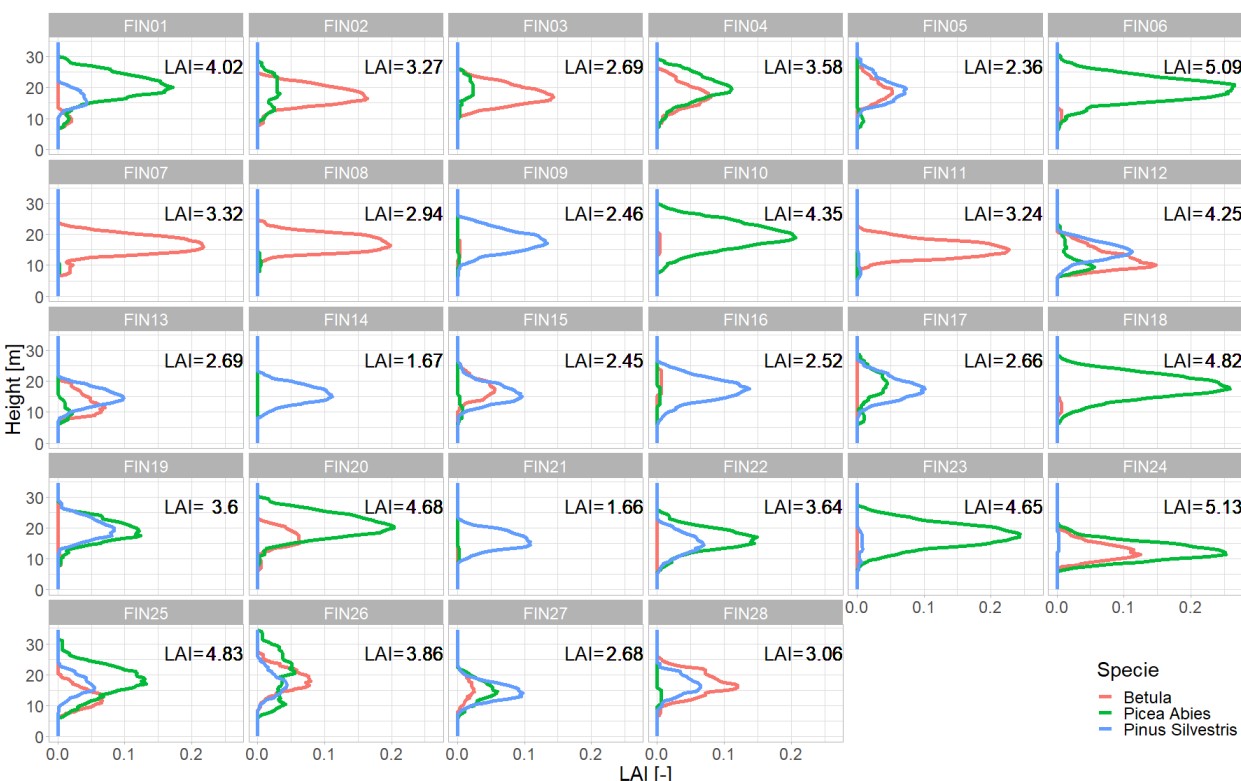

**Figure A5.** LAI Profiles for each forest stand. Shown are the LAI values (x-axis) in each height layer (y-axis) per species (red—Betula, green—Picea Abies, blue—Pinus Sylvestris). The colored lines show the LAI for a particular species (sum of all trees of the species in the plot). Therefore, the sum of all the lines gives the LAI profile of all trees in the plot.

## Appendix B. Additional Information on the Result Section

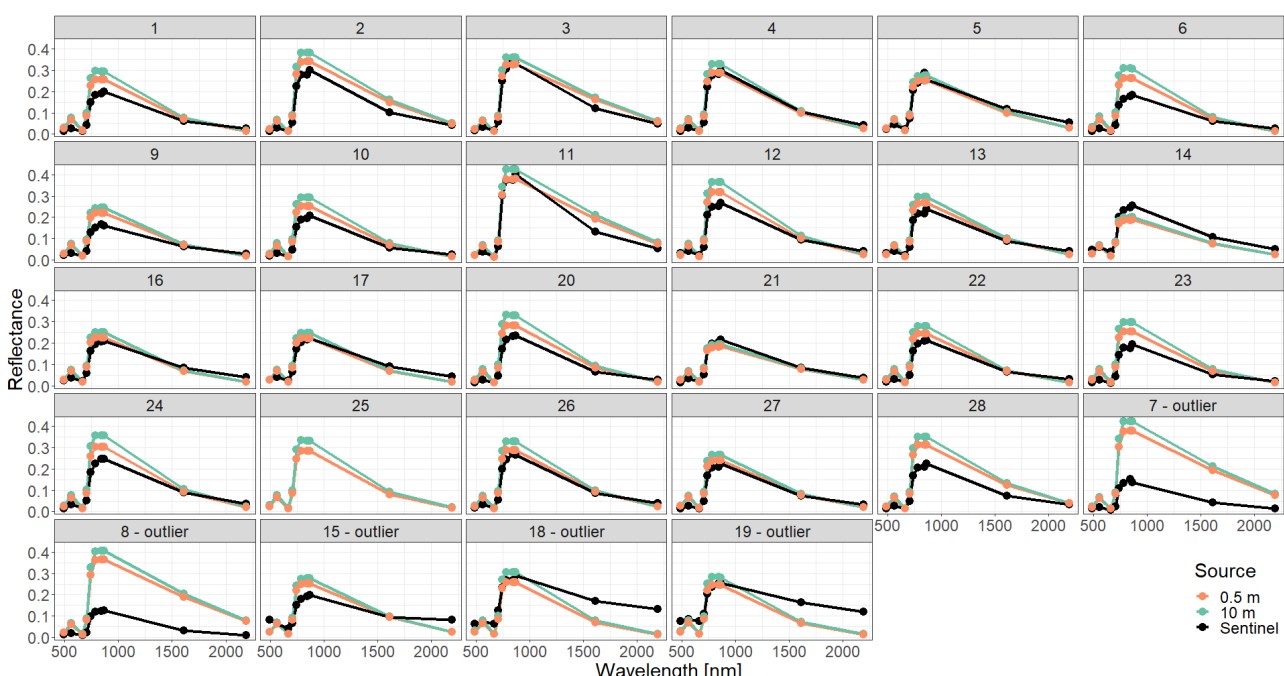

**Figure A6.** Comparison of simulated reflectance spectra with Sentinel measurements assuming a simple forest representation using different descriptions of the vertical forest structure (0.5 m or 10 m height layers). The Sentinel 2 spectrum of plot 25 is not provided by [28].

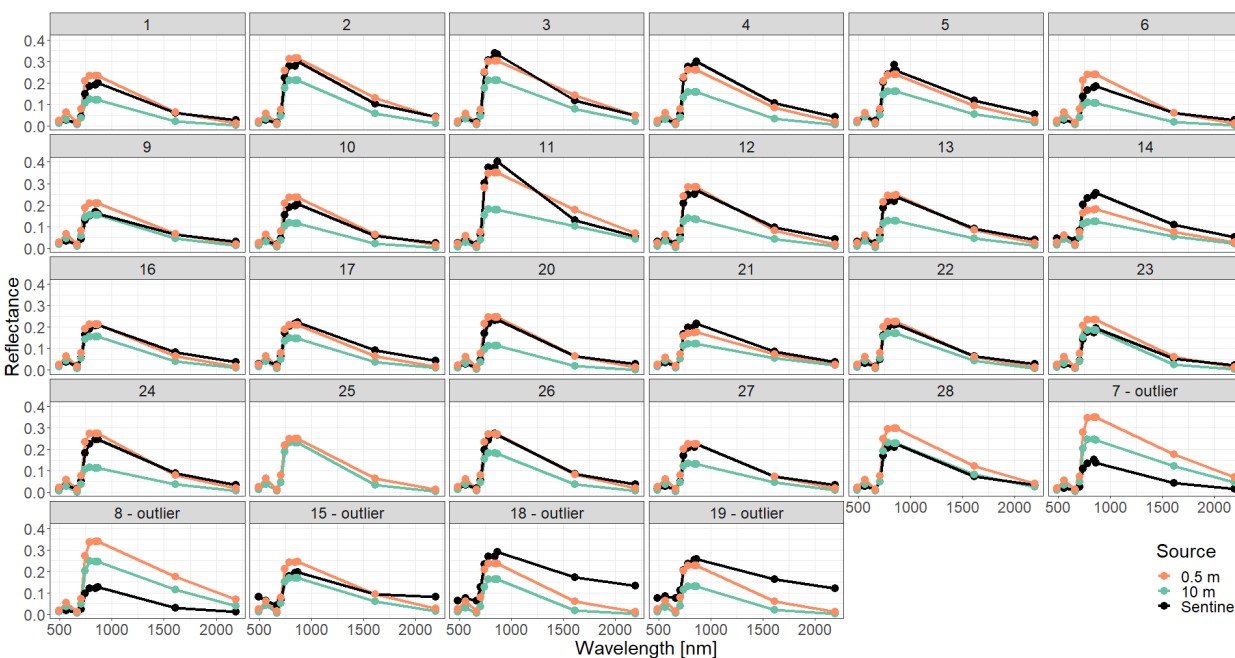

**Figure A7.** Comparison of simulated reflectance spectra with Sentinel measurements assuming a detailed forest representation using different descriptions of the vertical forest structure (0.5 m or 10 m height layers).

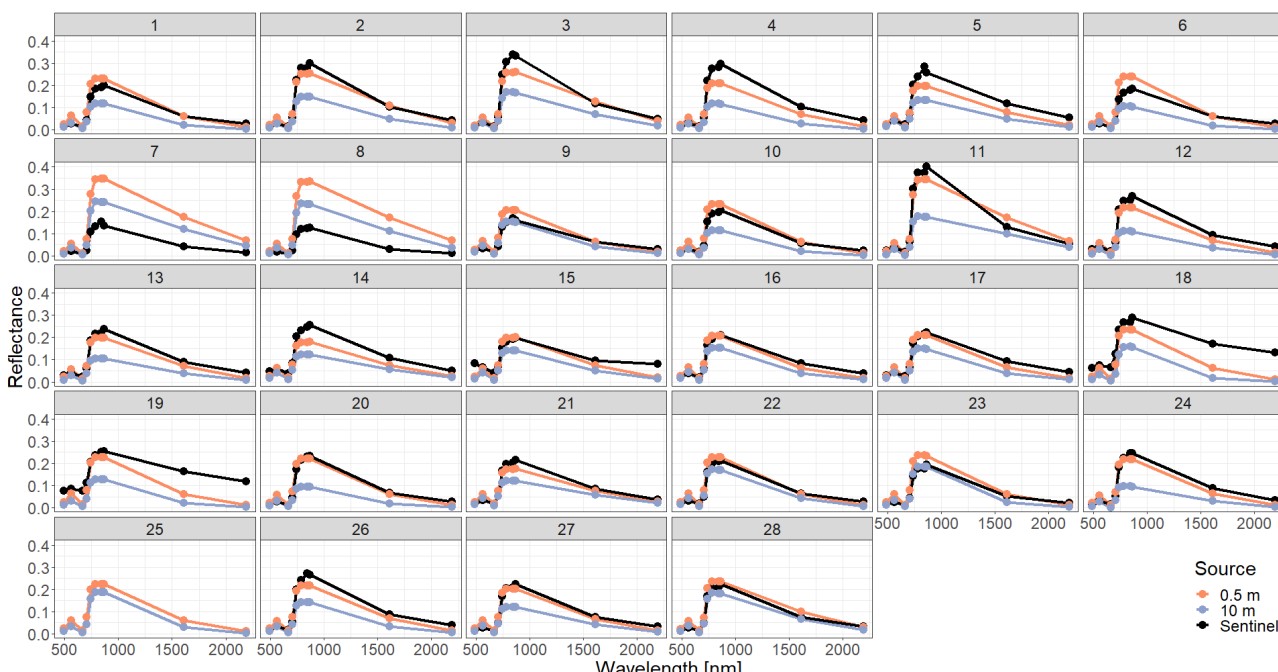

**Figure A8.** Comparison of simulated reflectance spectra with Sentinel measurements assuming a spectra-averaged forest representation using different descriptions of the vertical forest structure (0.5 m or 10 m height layers).

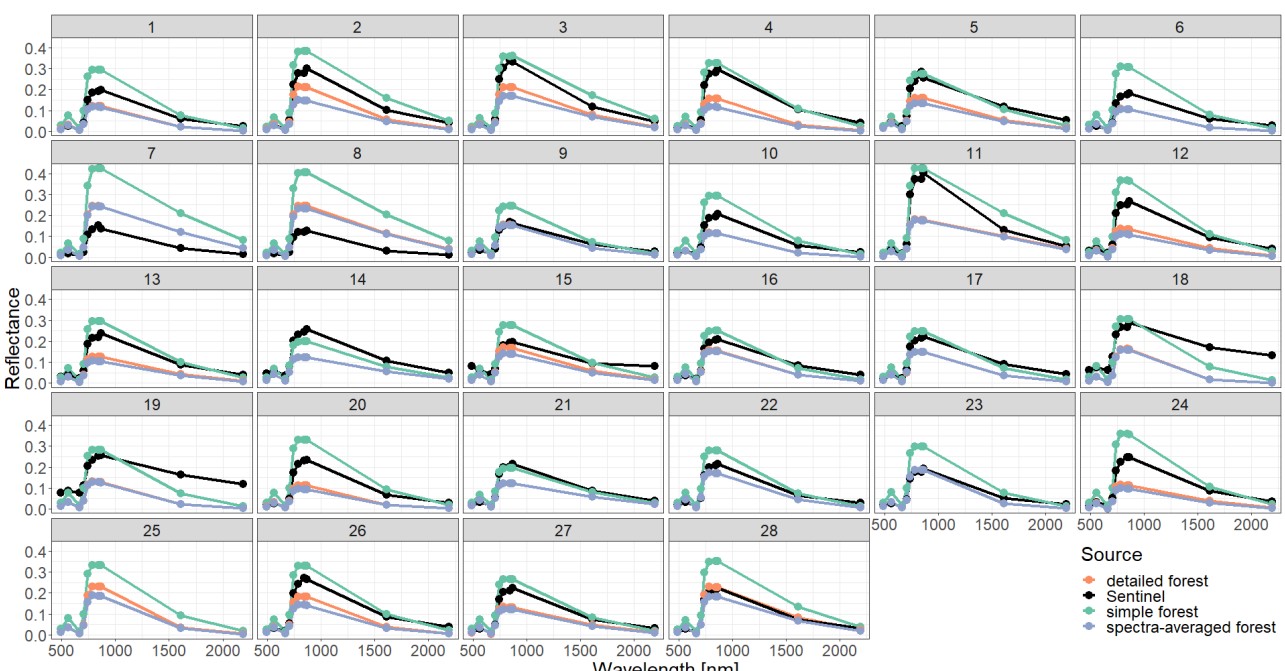

**Figure A9.** Comparison of simulated reflectance spectra with Sentinel measurements assuming different types of forest representations (simple, detailed and specta-averaged) and using 10 m height layers for the description of vertical forest structure.

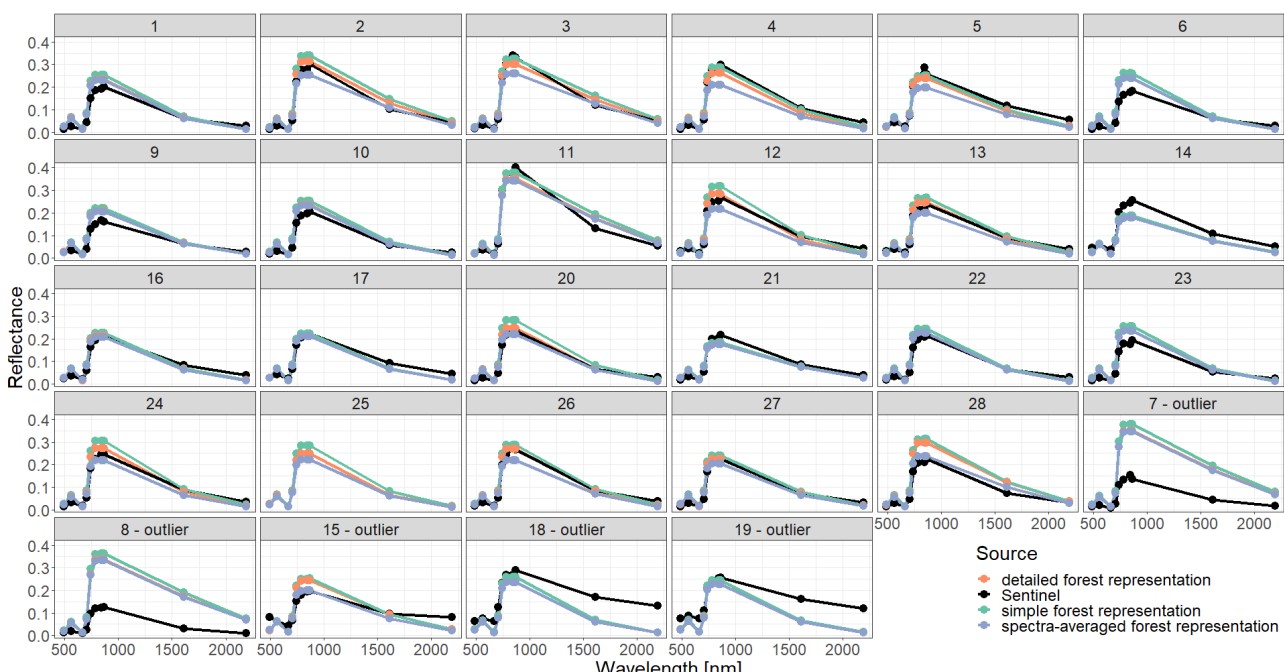

**Figure A10.** Comparison of simulated reflectance spectra with Sentinel measurements assuming different types of forest representations (simple, detailed, spectra-averaged) and using 0.5 m height layers for the description of vertical forest structure.

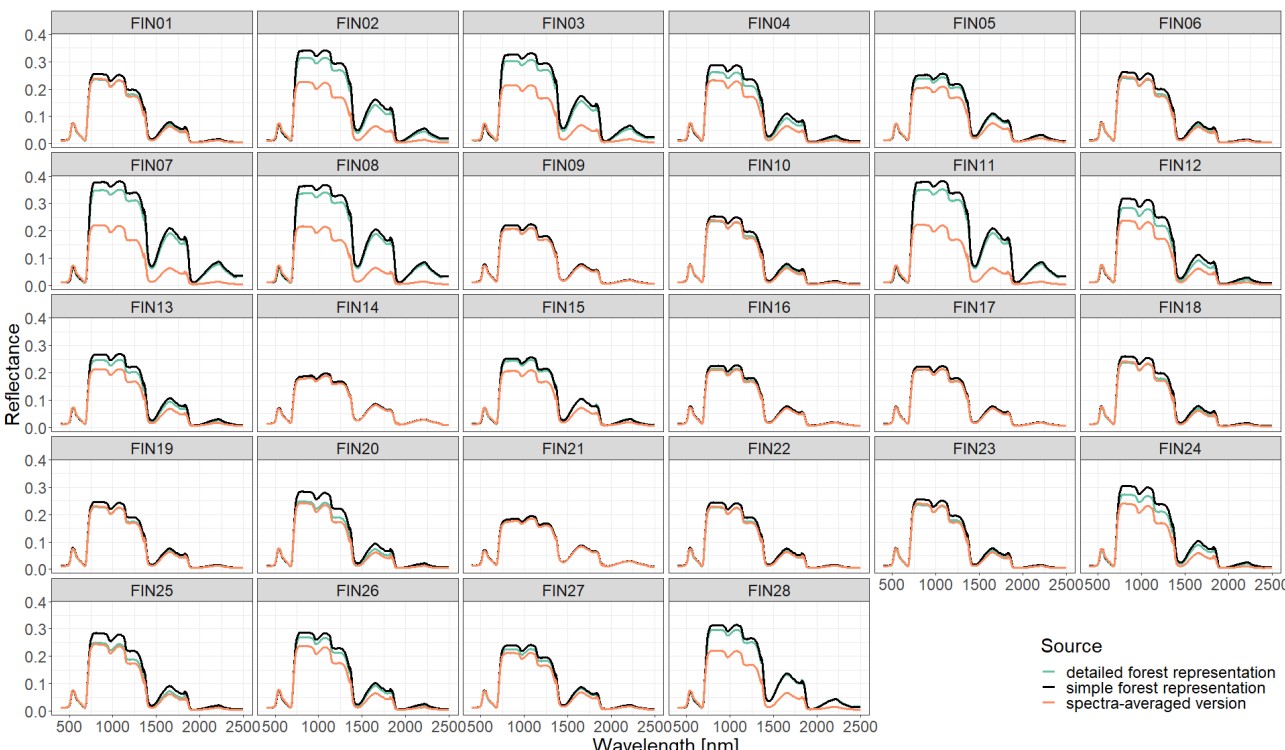

**Figure A11.** Comparison of simulated reflectance spectra assuming different types of forest representations (simple, detailed, spectra averaged) and using 0.5 m height layers for the description of vertical forest structure. The resolution of the simulated reflectance wavelengths is 1 nm.

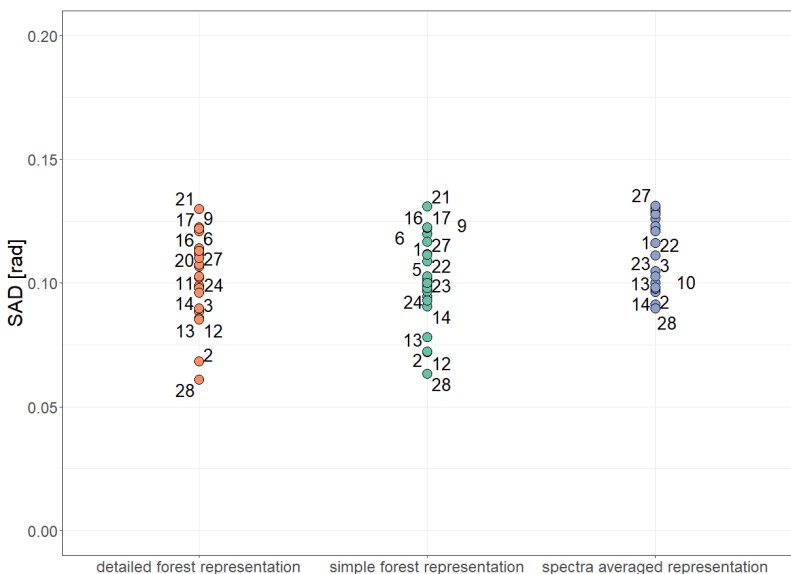

**Figure A12.** Comparison of simulated reflectance spectra and measured reflectance spectra using a distance index (Spectral Angle Distance, 0 rad: identical, $\frac{\pi}{2}$ rad: different). Results are shown for 28 forest plots (dots) and different forest representations. Comparison has been conducted for 10 wavebands of Sentinel-2.

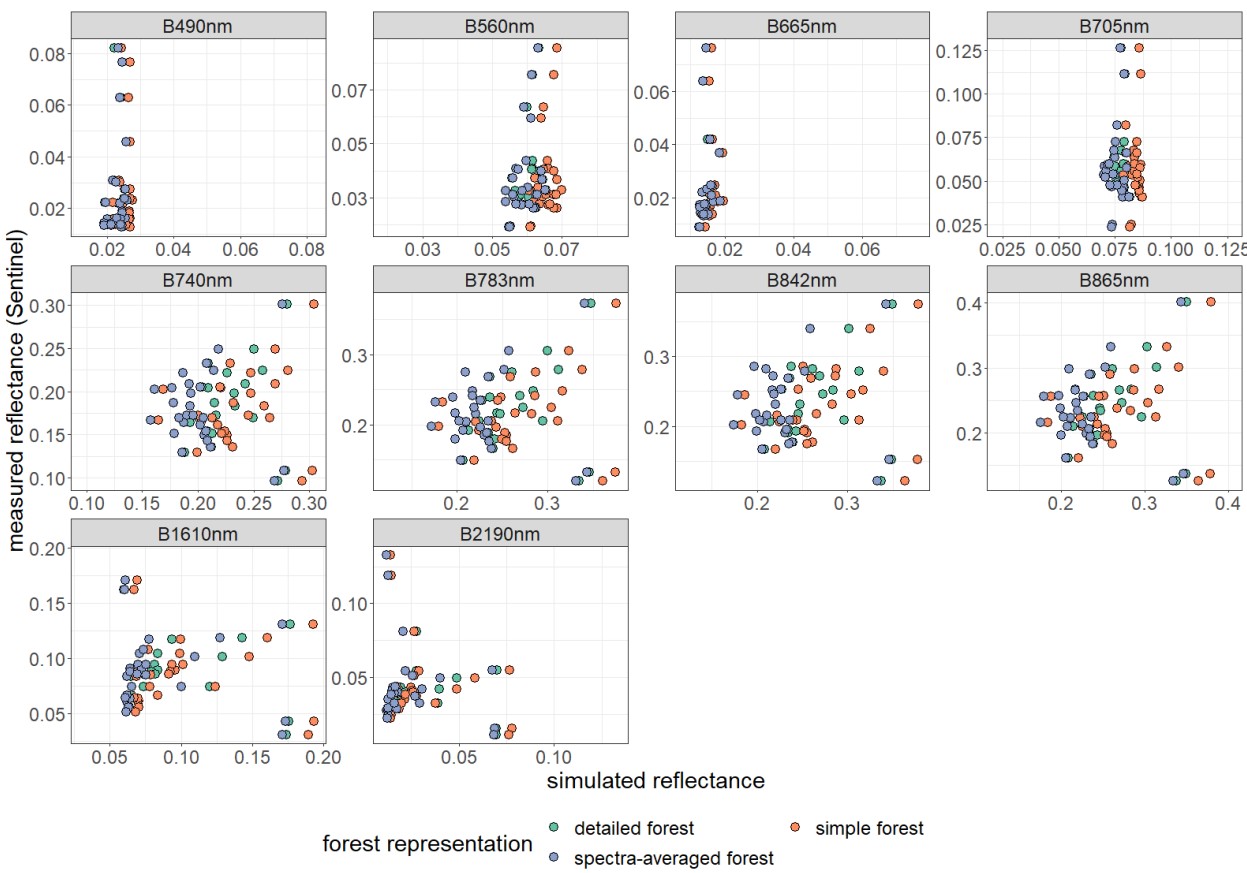

**Figure A13.** Comparison of simulated and measured reflectance of 28 forest stands (dots) and different forest representations (indicated by colors). Please note that the scales used for the illustration of measured and simulated reflectance differs for each band. Reflectance has been averaged for 10 wavebands (described by centered wavelengths; for more information on wavebands see Table A3).

## Appendix C. Analysis of Selected Forest Stands (Outliers)

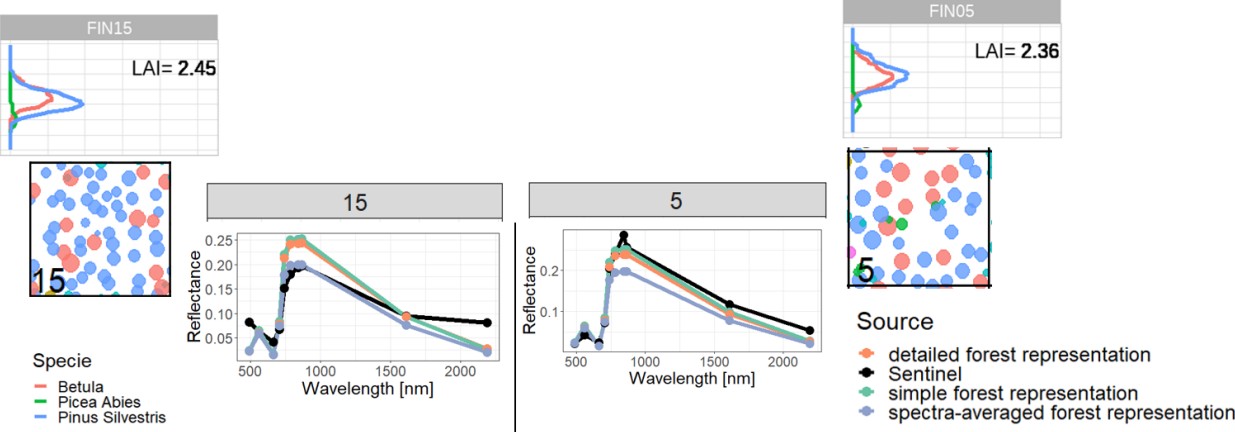

**Figure A14.** Comparison of reflectance spectra and additional information of forest stand 15 (classified as outlier) with forest stand 5. We compare forest properties of an outlier (left side) with forest properties of a forest with similar attributes, which is not an outlier (right side). Therefore, we compare the LAI profile (outer sides top), the reflectance spectra (in the middle) and the species composition (outer sides bottom). Despite the similar LAI distribution and species composition we obtained different Sentinel-measurements for reflectance, but similar simulated reflectance.

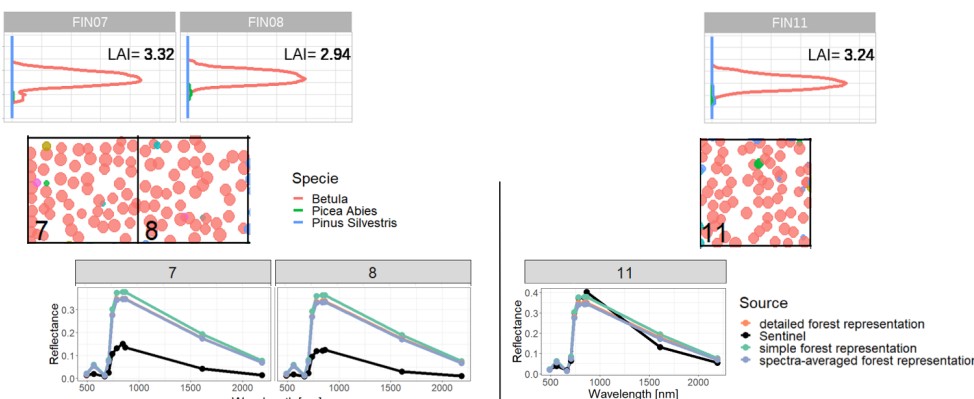

**Figure A15.** Comparison of reflectance spectra and additional information of forest stands 7 and 8 (classified as outliers) with forest stand 11. We compare forest properties of two outliers (left side) with forest properties of a forest with similar attributes, which is not an outlier (right side). Therefore, we compare the LAI profile (outer sides top), the reflectance spectra (in the middle) and the species composition (outer sides bottom). Despite the similar LAI distribution and species composition, we obtained different Sentinel measurements for reflectance, but similar simulated reflectance.

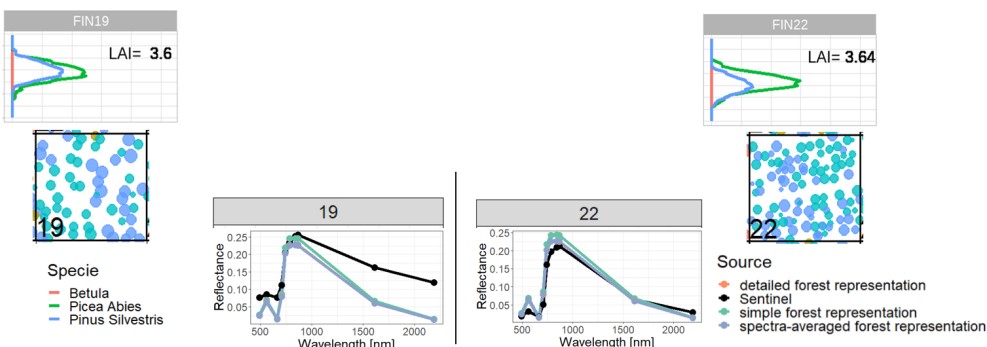

**Figure A16.** Comparison of reflectance spectra and additional information of forest stand 19 (classified as outlier) with forest stand 22. We compare forest properties of an outlier (left side) with forest properties of a forest with similar attributes, which is not an outlier (right side). Therefore, we compare the LAI profile (outer sides top), the reflectance spectra (in the middle) and the species composition (outer sides bottom). Despite the similar LAI distribution and species composition, we obtained different Sentinel measurements for reflectance, but similar simulated reflectance.

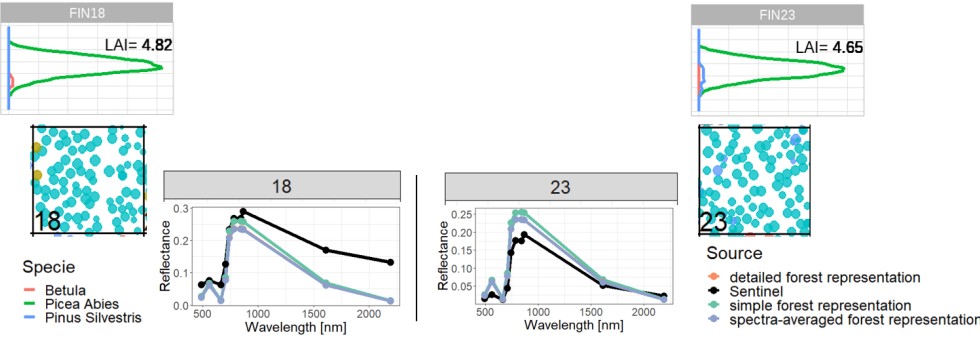

**Figure A17.** Comparison of reflectance spectra and additional information of forest stand 18 (classified as outlier) with forest stand 23. We compare forest properties of an outlier (left side) with forest properties of a forest with similar attributes, which is not an outlier (right side). Therefore, we compare the LAI profile (outer sides top), the reflectance spectra (in the middle) and the species composition (outer sides bottom). Despite the similar LAI distribution and species composition, we obtained different Sentinel measurements for reflectance, but similar simulated reflectance.

## Appendix D. Analysis of LAI and Additional Indices

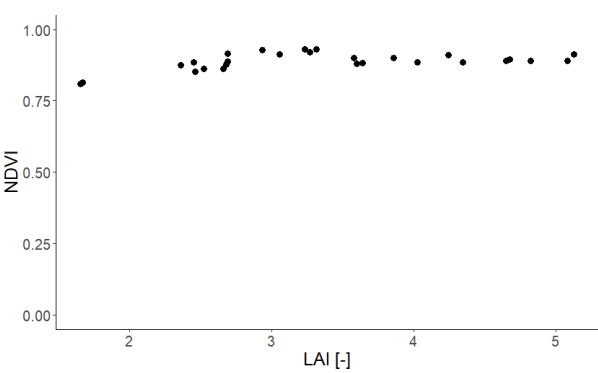

**Figure A18.** Relationship between LAI (x-axis, field data) and NDVI (y-axis, Sentinel-2 measurements) of 28 Finland forest stands.

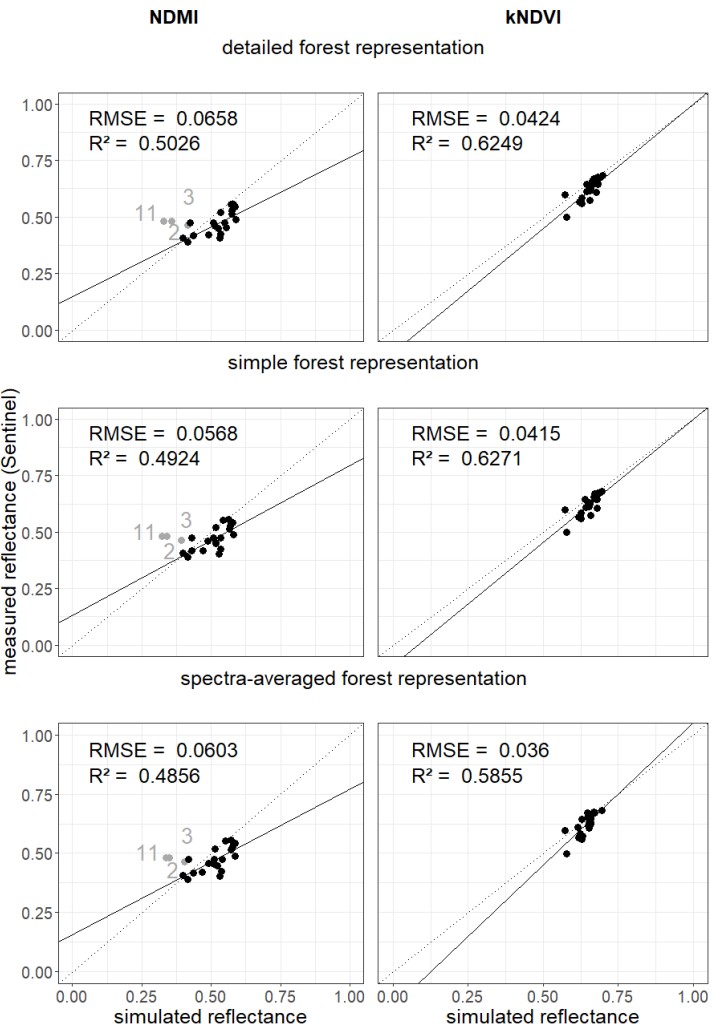

**Figure A19.** Comparison of vegetation indices for 28 forest stands in Finland. The vegetation indices (NDMI left, kNDVI right) for the measured reflectance spectra in the different wavebands for the simulated reflectance spectra (x-axis) and for the satellite measurements (y-axis). In each row, a different forest representation is assumed (1. detailed forest representation, 2. simple forest representation, 3. spectra-averaged forest representation; more information about the cases in Section 2.3). Each point represents a forest stand in Finland (gray points indicate outliers that are not used to calculate the RMSE and $R^2$—see Appendix Figures A14–A17).

**Table A4.** Analysis of vegetation indices for different forest representations. The mean spectral angle distance is calculated as average of the SAD of 28 forest stands for each forest representation (height layer size = 0.5 m). More details about the vegetation indices and spectral angle distance can be found in Section 2.4, Figures 5 and A19.

|  | **Simple Forest** | **Detailed Forest** | **Spectra Averaged Forest** |
|---|---|---|---|
| **NDVI** | | | |
| $R^2$ | 0.63 | 0.63 | 0.59 |
| bias $R^2$ | $-0.086$ | $-0.097$ | $-0.177$ |
| RMSE | 0.04 | 0.04 | 0.033 |
| MAE | 0.033 | 0.034 | 0.027 |
| **EVI** | | | |
| $R^2$ | 0.43 | 0.45 | 0.25 |
| bias $R^2$ | 0.086 | 0.059 | 0.116 |
| RMSE | 0.107 | 0.081 | 0.074 |
| MAE | 0.092 | 0.069 | 0.062 |
| **MSI** | | | |
| $R^2$ | 0.49 | 0.49 | 0.47 |
| bias $R^2$ | 0.141 | 0.162 | 0.164 |
| RMSE | 0.051 | 0.059 | 0.054 |
| MAE | 0.041 | 0.05 | 0.043 |
| **NDMI** | | | |
| $R^2$ | 0.49 | 0.5 | 0.49 |
| bias $R^2$ | 0.133 | 0.15 | 0.156 |
| RMSE | 0.057 | 0.066 | 0.060 |
| MAE | 0.046 | 0.056 | 0.049 |
| **kNDVI** | | | |
| $R^2$ | 0.63 | 0.62 | 0.59 |
| bias $R^2$ | $-0.089$ | $-0.098$ | $-0.15$ |
| RMSE | 0.041 | 0.042 | 0.036 |
| MAE | 0.034 | 0.035 | 0.029 |
| mean SAD | 0.101 | 0.103 | 0.113 |

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
