# Peer review of "A New Approach Combining a Multilayer Radiative Transfer Model with an Individual-Based Forest Model: Application to Boreal Forests in Finland"

_remotesensing, doi:10.3390/rs15123078_

Round 1
Reviewer 1 Report
Comments
Major concern:
The most important point is if the values/ approach is inconsistent with Sentinel-2 data, how is the approach helpful when someone wants to apply it to satellite imagery (Sentinel-2 data) and what additional criteria are required to identify the applicable and not-applicable areas? Moreover, the study indicated significant differences depending on the diversity of each plot, which is highly inconsistent and specific to a limited region or forest type. The actual reason for differences is not identified or addressed.
The seasonal variabilities are unavailable: in field data collection, modelling and Sentinel-2 data.
The authors have not used a sample of the data for modelling and the rest for validation. Therefore, the scope of the study (approach and results) is limited to the study sites only.
Other comments:
The Introduction section can be improved by describing the overall RTM concept and adding factors regulating vegetation reflectance, scattering mechanism, leaf and plant structure, leaf foliar traits, etc.
The basic description of the FORMIND model added in the Methodology section can be shifted to the Introduction section.
A study area section can be added with basic details of forest type, dominant species, environmental conditions, etc.
Fig. 1: Add the unit of crown diameter. The color dot size in the species legend needs to be increased. It is unclear what sequence is followed in arranging the plots.
At what height the stem diameter is measured? Does the field data shown here include all the trees, scrubs/ shrubs, and grasses observed in the plots? Are there any seasonal grasses available in these plots?
It would be great if a study map is added with sample plot locations.
FORMIND model description: briefly mention what are the model’s input/ independent variables.
What is the purpose of reconstructing 2015 data instead of using 2012 (calibration) and simulating the variables for 2017, and validating with 2017 observation?
What were the other inputs of FORMIND used here? It will be useful seeing the allometric equations used to simulate the tree height, crown diameter and LAI using stem diameter and tree species information. Was the plot size (ground data) 30 m × 30 m? How to ensure the simulation accuracy? It would have been great if there was ground data on tree height, crown diameter and LAI values.
“We then compared the calculated reflectance spectra with remote sensing observation using atmosphere-corrected Sentinel-2 measurements [43] from 2015. For the simulation of the reflectance spectra, information on observation geometries (sun and observer, in terms of zenith and azimuth) for each forest stand is available. The measured reflectances (Sentinel-2) were used for comparison with the simulated reflectances.” – is it a part of section 2.2?
PROSPECT-D – is it a part of mScope or a stand-alone model? Not introduced properly.
Are the parameters used in PROSPECT-D, i.e., leaf structure, chlorophyll, etc., available for this region (biogeographic) or generalized values?
“each height layer has a layer height, here 0.5 m” – Unclear. Does it represent the simulation at every 0.5 m?
“it was assumed to be spherical for all species” – is there an option available to apply variable distribution depending on the species?
“we averaged the simulated reflectance values in the band range” – does it indicates the average from a hyperspectral spectrum to broadband spectra?
“The even-aged forest stands no. 17,” – this information (for each plot) is unavailable for reference.
“First, we analyzed the reflectance of even-aged forests, where the RTM used 10 m height layers” – why 10 m height? Are there other height-based values available? It seems that the entire approach adopted variable height information for each plot, then why particular height-based values are mentioned?
“Figure 3. Reflectance spectra for detailed and simplified forest representation by using layers of 10 m height.” “Figure 4. Reflectance spectra for detailed and simplified forest representation by using standard layers of 0.5 m height.” – it is unclear why these two height layers are studied. Why spectra averaged forest is not added in Figure 3, if these two figures compare the height variation?
Figure A2. Comparison of reflectance spectra for different layer heights with simple forest representation; Figure A3. Comparison of reflectance spectra for different layer heights with detailed forest representation. – Is it unavailable for the spectra-averaged forest?
Figure A4. Comparison of reflectance spectra for different concepts of forest representations with layer height of 10 m. Figure A5. Comparison of reflectance spectra for different concepts of forest representations with layer height of 0.5 m. – why the spectra-averaged forest is not added for 10 m height?
“The lowest reflectances were produced with the spectra-averaged forest representation (in particular for forest stand no. 5).” – authors have not mentioned an important result that the spectra-averaged forest in plot 5 has significantly underestimated the NIR reflectance. This will significantly underestimate the vegetation indices value.
Figure 5: The RMSE values for MSI is quite high. It is unclear why plot 15 is not considered in the MSI comparison (how the outliers are decided?). Add bias and MAE values. A table with comparative statistics for simple, detailed and spectra-averaged curves.
Is it possible to add plots for reflectance values similar to figure 5?
Author Response
Dear Ms. Kristie Shen, dear Ms. Mabel Wang and dear reviewers,
Thank you for your time and effort in the reviewing process of our manuscript (remotesensing-2390304). We are grateful for the opportunity to submit a revised version of our manuscript and we would further like to thank the reviewers for the constructive feedback and helpful suggestions.
We thoroughly considered all comments of the reviewers. In particular we revised the description of mScope as well as of FORMIND and we added necessary details in order to improve the overall clarity of the study design. We added additional analysis to improve the presentation of the comparison of resulting reflectance and indices. The discussion was expanded to include some additional limitations. Once more, many thanks to the reviewers for their comments.
Below you find a line-by-line response to the issues raised by the reviewers. The replies are written in italic. The given line numbers are related to the new version of the manuscript. The track changes version can be found attached.
On behalf of the authors,
Hans Henniger
PS: Additionally, we want to inform you that the author of the study which provides the leaf parameters (Kothari et al., 2023 [48]) contacted us to share a mistake he made. In consequence we had to adjust the parameter describing the leaf water content and recalculate our results. For the most part, this only has an influence on waveband 11 (centred wavelength of 1610) and 12 (centred wavelength of 2190) as well as the water indices (MSI and NDMI).
Major concern:
The most important point is if the values/ approach is inconsistent with Sentinel-2 data, how is the approach helpful when someone wants to apply it to satellite imagery (Sentinel-2 data) and what additional criteria are required to identify the applicable and not-applicable areas? Moreover, the study indicated significant differences depending on the diversity of each plot, which is highly inconsistent and specific to a limited region or forest type. The actual reason for differences is not identified or addressed.
- Reply: Thank you very much for the comment. We added text in the discussion to point out the limits of our approach (L318ff). We agree that for the investigated forest stands it is no possible to extract clear information on the question which type of forest stand is more difficult to analyze with our approach. We added text to make clear why we consider five forest stands as outliers and why we exclude them from parts of our analysis (Figure 5 and L333ff). For these forest stands we got different Sentinel measurements even if the forest stands have similar species mixtures and LAI distribution. Other factors, that are missing in the field data (like understory) may lead to these differences in Sentinel measurements. Additionally, we added an index for direct comparison of the measured and simulated reflectances (SAD in Figure A10).
The seasonal variabilities are unavailable: in field data collection, modelling and Sentinel-2 data.
- Reply: Thank you very much for the comment. We added this information in the text (L129, L 144). The field data was collected in summer (July/August) and the Sentinel images are also chosen from August 2015.
The authors have not used a sample of the data for modelling and the rest for validation. Therefore, the scope of the study (approach and results) is limited to the study sites only.
- Reply: Thanks for the interesting comment. For this study we used two well established models (FORMIND and mScope) which have been already validated in various studies. The Idea of this study was to use general parameters about forest stands and leaf attributes for reproducing the reflectance of forests. We combined the models and used well established parameters (out of inventories, field studies) and made a first attempt to apply the developed approach to forests in Finland.
Other comments:
The Introduction section can be improved by describing the overall RTM concept and adding factors regulating vegetation reflectance, scattering mechanism, leaf and plant structure, leaf foliar traits, etc.
- Reply: Thank you for your suggestion. We added additional descriptions of the RTM approach (L64ff).
The basic description of the FORMIND model added in the Methodology section can be shifted to the Introduction section. A study area section can be added with basic details of forest type, dominant species, environmental conditions, etc.
- Reply: Thanks for the suggestion. We added some general information about the FORMIND model in the introduction (L54ff) Additionally, we split section 2.1. in a section including information on the study site and a section with more information on the FORMIND model (L125ff and L147ff).
Fig. 1: Add the unit of crown diameter. The color dot size in the species legend needs to be increased. It is unclear what sequence is followed in arranging the plots. environmental conditions, etc.
- Reply: Thanks for noticing. We added the information and adjusted the dot size (Fig. 1). The sequence of plot numbers has historical reasons (related to the FORMIND architecture).
At what height the stem diameter is measured? Does the field data shown here include all the trees, scrubs/ shrubs, and grasses observed in the plots? Are there any seasonal grasses available in these plots?
- Reply: Thank you for this question. We added information about using the dbh (diameter on breast height) (L159ff). The field data provided no information on other plants than trees. We added that information (L. 132). But indeed, it would be interesting and important to have additional information on grasses in future studies (L 348).
It would be great if a study map is added with sample plot locations.
- Reply: Thank you. We added a map in the Appendix (Figure A1).
FORMIND model description: briefly mention what are the model’s input/ independent variables.
- Reply: Thanks for the suggestion. We split section 2.1. and added information about input variables and processes of the forest model (L. 147ff). We also added information, on which parameterization we refer and where to find it (L. 196).
What is the purpose of reconstructing 2015 data instead of using 2012 (calibration) and simulating the variables for 2017, and validating with 2017 observation?
- Reply: Thank you for the comment. We added text to explain this (L 139ff). We used data that was already used in other studies. Here we used the Sentinel data of 2015 from the publication of Ma et al. (2019) and derived an estimate of tree stem diameters for 2015 from the available forest inventories (2012, 2015).
What were the other inputs of FORMIND used here? It will be useful seeing the allometric equations used to simulate the tree height, crown diameter and LAI using stem diameter and tree species information. Was the plot size (ground data) 30 m × 30 m? How to ensure the simulation accuracy? It would have been great if there was ground data on tree height, crown diameter and LAI values.
- Reply: Thanks for pointing out and the suggestion for an additional Figure. We added figures and tables to the Appendix to describe the mentioned allometries (Figure A3, Table A2). The plot size was 30 m x 30m in field inventories. We used the same plot size for forest reconstruction with FORMIND (L. 130, L. 165, L. 201).
“We then compared the calculated reflectance spectra with remote sensing observation using atmosphere-corrected Sentinel-2 measurements [43] from 2015. For the simulation of the reflectance spectra, information on observation geometries (sun and observer, in terms of zenith and azimuth) for each forest stand is available. The measured reflectances (Sentinel-2) were used for comparison with the simulated reflectances.” – is it a part of section 2.2?
- Reply: Thank you. We changed this text to focus it more on the description of the source of our data (L. 146).
PROSPECT-D – is it a part of mScope or a stand-alone model? Not introduced properly.Are the parameters used in PROSPECT-D, i.e., leaf structure, chlorophyll, etc., available for this region (biogeographic) or generalized values?
- Reply: Thanks for the question. We added more information about Prospect D and the parameters we used (L. 182, L. 208, L 192). The parameters have been taken from a data base and represent generalized values (see Table A1). Values from this region are unfortunately not available. We averaged the values for the leaf attributes (for different species) from the data base and used averaged values.
“each height layer has a layer height, here 0.5 m” – Unclear. Does it represent the simulation at every 0.5 m?
- Reply: Thank you for this helpful comment. We added more information to the text to make this clearer (L 201, L 274). MScope uses height layers to describe the heterogenous structure of forests. So, we had to extract the required information from the forest model considering the height layers. The RTM uses input information (e.g. species mix, LAI) for each 0.5 m height layer to calculate the reflection for a forest.
“it was assumed to be spherical for all species” – is there an option available to apply variable distribution depending on the species?
- Reply: Thank you for this comment. We added more information about the leaf angle distribution (L. 211). The spherical distribution is often used in studies, but also other distributions are possible.
“we averaged the simulated reflectance values in the band range” – does it indicates the average from a hyperspectral spectrum to broadband spectra?
- Reply: Thank you for this question. We added an example of how we average reflectance values in wavebands (L. 243). With our approach we can simulate reflectance for wavelengths between 400 and 2500 nm with a resolution of 1 nm. Sentinel 2 provides reflectance values for 13 bands (Table A3), for which three of them were used for the calibration of the Sentinel measurements.
“The even-aged forest stands no. 17,” – this information (for each plot) is unavailable for reference.
- Reply: Thank you for this input. We added a table in the Appendix with additional information about all investigated forest plots. (Table A2)
“First, we analyzed the reflectance of even-aged forests, where the RTM used 10 m height layers” – why 10 m height? Are there other height-based values available? It seems that the entire approach adopted variable height information for each plot, then why particular height-based values are mentioned?
- Reply: Thanks for the question. We added more information on height layers (201ff). The analysis using 10 m height layers is an approach with less complexity (resulting in faster computation) and less information on the heterogeneity of the forests. The analysis shows that a certain level of complexity is necessary to achieve good results with the mScope model, even if it implies higher computational costs.
“Figure 3. Reflectance spectra for detailed and simplified forest representation by using layers of 10 m height.” “Figure 4. Reflectance spectra for detailed and simplified forest representation by using standard layers of 0.5 m height.” – it is unclear why these two height layers are studied. Why spectra averaged forest is not added in Figure 3, if these two figures compare the height variation?
Figure A2. Comparison of reflectance spectra for different layer heights with simple forest representation; Figure A3. Comparison of reflectance spectra for different layer heights with detailed forest representation. – Is it unavailable for the spectra-averaged forest?
Figure A4. Comparison of reflectance spectra for different concepts of forest representations with layer height of 10 m. Figure A5. Comparison of reflectance spectra for different concepts of forest representations with layer height of 0.5 m. – why the spectra-averaged forest is not added for 10 m height?
- Reply: Thank you very much for the comment. As mentioned in the reply above, we added more information on the height layers. We believe that the results for the spectra-averaged case and 10 m height layers gives only minor additional insights (compared to the case with 0.5 m height layers). So, we did not include this additional analysis to the Appendix.
“The lowest reflectances were produced with the spectra-averaged forest representation (in particular for forest stand no. 5).” – authors have not mentioned an important result that the spectra-averaged forest in plot 5 has significantly underestimated the NIR reflectance. This will significantly underestimate the vegetation indices value.
- Reply: Thanks for pointing out. We added the requested information. (L292)
Figure 5: The RMSE values for MSI is quite high. It is unclear why plot 15 is not considered in the MSI comparison (how the outliers are decided?). Add bias and MAE values. A table with comparative statistics for simple, detailed and spectra-averaged curves.
- Reply: Thanks for pointing this out and for the suggestion of an additional table. We added a table (Table A4) with results on the analysis of vegetation indices (including bias, MAE and a distance index – spectral angle distance SAD). In the analysis of the outliers (plot 15 is included in this analysis in Figure A12) we showed that Sentinel measurements (e.g. for plot 15 and plot 5) are different even if the species mix and the forest structure are more or less similar. Our conclusion is that other factors, which are not included in the field data, are responsible for these differences. That’s why we didn’t included forest nr. 15 for the calculation of R² for the MSI and other indices (Figure A5).
Is it possible to add plots for reflectance values similar to figure 5?
- Reply: Thanks for the suggestion. We added a figure with the comparison of the reflectance values for Sentinel 2 wavebands and the different types of forest representation. (Figure A11)

Reviewer 2 Report
The article "A new approach combining a multilayer radiative transfer model with an individual based forest model: Application to boreal forests in Finland". The authors proposes a method that couples the RTM mScope with the individual-based forest model FORMIND, and compares the simulation output with Sentinel-2 data using Finnish forests as an example. The research has some novelty and the method helps to understand the relationship between forest reflectance and forest properties. However, there are still some shortcomings that need improvement:
1. It is suggested to add comparisons with other methods to demonstrate the advantages of this method.
2. In the "3. Results" section, it is recommended to use evaluation indicators to determine which type of forest simulation spectra is more similar to satellite spectra, such as the Spectral Angle Distance (SAD).
3. Improvements are suggested for Figure 5, as the numbers in the figure overlap.
4. The reason why the simple and detailed forest representation is better than spectral average forest representation should be explained.
5. It is suggested to share the model code.
6. A critical view of the potential drawbacks of this method is needed.
Minor editing of English language required
Author Response
Dear Ms. Kristie Shen, dear Ms. Mabel Wang and dear reviewers,
Thank you for your time and effort in the reviewing process of our manuscript (remotesensing-2390304). We are grateful for the opportunity to submit a revised version of our manuscript and we would further like to thank the reviewers for the constructive feedback and helpful suggestions.
We thoroughly considered all comments of the reviewers. In particular we revised the description of mScope as well as of FORMIND and we added necessary details in order to improve the overall clarity of the study design. We added additional analysis to improve the presentation of the comparison of resulting reflectance and indices. The discussion was expanded to include some additional limitations. Once more, many thanks to the reviewers for their comments.
Below you find a line-by-line response to the issues raised by the reviewers. The replies are written in italic. The given line numbers are related to the new version of the manuscript. The track changes version can be found attached.
On behalf of the authors,
Hans Henniger
PS: Additionally, we want to inform you that the author of the study which provides the leaf parameters (Kothari et al., 2023 [48]) contacted us to share a mistake he made. In consequence we had to adjust the parameter describing the leaf water content and recalculate our results. For the most part, this only has an influence on waveband 11 (centred wavelength of 1610) and 12 (centred wavelength of 2190) as well as the water indices (MSI and NDMI).
Review:
The article "A new approach combining a multilayer radiative transfer model with an individual based forest model: Application to boreal forests in Finland". The authors proposes a method that couples the RTM mScope with the individual-based forest model FORMIND, and compares the simulation output with Sentinel-2 data using Finnish forests as an example. The research has some novelty and the method helps to understand the relationship between forest reflectance and forest properties. However, there are still some shortcomings that need improvement:
- It is suggested to add comparisons with other methods to demonstrate the advantages of this method.
- Reply: Thank you very much for the comment. We are not sure if we understand the comment properly. In this study we already tried to compare different methods of using the approach of FORMIND with mScope. We investigated different methods of the implementation of forest structure (different forest representations) for different resolutions (different size of height layers) and analyzed their influence on simulated forest reflection in comparison with Sentinel measurements. Comparison with results from other forest and radiative transfer models for this region is beyond the scope of this study.
- In the "3. Results" section, it is recommended to use evaluation indicators to determine which type of forest simulation spectra is more similar to satellite spectra, such as the Spectral Angle Distance (SAD).
- Reply: Thanks for pointing out and the suggestion for an additional indicator. We added a table with the results for all plots (including bias and SAD) and added a figure with the analysis of SAD. (Figure A10, Table A4)
- Improvements are suggested for Figure 5, as the numbers in the figure overlap.
- Reply: Thanks for noticing. We excluded the outlier plots from the figure to avoid misunderstanding and overlapping. Weak results for the calculation of MSI has been obtained for plots with main species of Betula. We marked them (gray colour) and excluded them from the calculation of R².
- The reason why the simple and detailed forest representation is better than spectral average forest representation should be explained.
- Reply: Thanks for the comment, this is an interesting question. We added text to our discussion to describe this in more detail (L329ff). The simple and detailed forest representations use an averaged leaf parameterization for each height layer (averaged using the LAI of the occurring species as weighting factor). In the spectrum-averaged version, we simulate each occurring species as a monoculture forest with detailed forest structure and afterwards average the resulting reflectance of the monoculture forest (using the LAI of the occurring species as weighting factor). Our results show that averaging the input leaf parameters provides better results than averaging the output reflectance of the approach, even though mScope is a nonlinear model.
- It is suggested to share the model code.
- Reply: Yes, we agree. The model code of FORMIND (without mScope) and mScope is available as open source. We want to publish our code for the coupling after the next two studies where we will apply it to different regions and a larger number of forest stands.
- A critical view of the potential drawbacks of this method is needed.
- Reply: Thanks for the suggestion. We added text and extended a paragraph in the discussion limitations of the approach (L. 322)

Round 2
Reviewer 1 Report
remotesensing-2390304
I am not convinced about the model parameterization and simulated results. There are major flaws and a lack of significance of the study.
Major Comments:
Figure 1: If the canopy density or total canopy closure is low, the sub-canopy or background land surface features significantly contributes to the pixel’s surface reflectance, which is not captured and represented in the study and may not be a true representation. In that case, the model parameterization and approach may lead to significant discrepancies applying elsewhere.
Figure A3: How realistic is the figure A3(iii), wherein the LAI doesn’t vary with stem diameter?
Figure A4: There are some LAI profiles which are completely 0 (zero) across the height. It means there is no canopy. Then how it justifies the presence of those trees in the plots?
Figure A4: The majority height/ maximum LAI is not at 10 m, rather mostly varies from 10 m to 15 m; then, why are the results focused on the 10 m model simulation, not the plot level majority height contribution given in Figure A5? Why is the 0.5 m simulation compared here instead of the 15 m and 20 m simulation? Sentinel-2 spectra is missing in plot 25.
The following responses with respect to queries (blue text) are not convincing:
“First, we analyzed the reflectance of even-aged forests, where the RTM used 10 m height layers” – why 10 m height? Are there other height-based values available? It seems that the entire approach adopted variable height information for each plot, then why particular height-based values are mentioned?
Reply: Thanks for the question. We added more information on height layers (201ff). The analysis using 10 m height layers is an approach with less complexity (resulting in faster computation) and less information on the heterogeneity of the forests. The analysis shows that a certain level of complexity is necessary to achieve good results with the mScope model, even if it implies higher computational costs.
“Figure 3. Reflectance spectra for detailed and simplified forest representation by using layers of 10 m height.” “Figure 4. Reflectance spectra for detailed and simplified forest representation by using standard layers of 0.5 m height.” – it is unclear why these two height layers are studied. Why spectra averaged forest is not added in Figure 3, if these two figures compare the height variation?
Figure A2. Comparison of reflectance spectra for different layer heights with simple forest representation; Figure A3. Comparison of reflectance spectra for different layer heights with detailed forest representation. – Is it unavailable for the spectra-averaged forest?
Figure A4. Comparison of reflectance spectra for different concepts of forest representations with layer height of 10 m. Figure A5. Comparison of reflectance spectra for different concepts of forest representations with layer height of 0.5 m. – why the spectra-averaged forest is not added for 10 m height?
Reply: Thank you very much for the comment. As mentioned in the reply above, we added more information on the height layers. We believe that the results for the spectra-averaged case and 10 m height layers gives only minor additional insights (compared to the case with 0.5 m height layers). So, we did not include this additional analysis to the Appendix. – If the change in height layers doesn’t add changes in reflectance, then there is a major drawback in the model and simulated results with varied LAI and height information. Moreover, authors must focus on the higher height layers, 15 m and 20 m, rather than 0.5 m and 10 m. 0.5 m representation may be more reliable/ valid for shrub/scrub-like life forms than trees.
Figure A11: This is the main story of the analysis, which shows the high biases in simulation compared to the observed (Sentinel-2) surface reflectance. It shows that the model is not sensitive enough to estimate the surface reflectance in the first four bands.
Figure A12: As the authors rightly pointed out, there are additional factors that control the surface reflectance in Sentinel-2, but they are not included and captured in the model simulation.
Figure A13 and A14: This shows that LAI and species composition doesn’t necessarily regulate the surface reflectance. The background reflectance from sub-canopy features and topography could significantly influence surface reflectance, and the current approach fails to capture that. Then what is the strength of the current study?
Table A4: The table doesn’t show any improvement in the model simulation with spectra averaged forest. Again, what is the significance of the study?
